# NLRP3-induced systemic inflammation controls the development of JAK2V617F mutant myeloproliferative neoplasms

Ruth-Miriam Koerber [1,2,3], Calvin Krollmann[1,3], Kevin Cieslak[1,3], Elisabeth Tregel[1,3], Maria L. Saenz [1,3], Tim H. Brümmendorf [4,5], Steffen Koschmieder [4,5], Martin Griesshammer[6], Ines Gütgemann[3,7], Conny K. Baldauf[8], Thomas Fischer[8], Peter Brossart[1,3], Carl Christian Kolbe[9], Eicke Latz [9,10], Dominik Wolf [11,12] ✉ & Lino L. Teichmann [1,3,12] ✉

The development of Philadelphia chromosome-negative classical myeloproliferative neoplasms (MPN) involves an inflammatory process that facilitates outgrowth of the malignant clone and correlates with clinical outcome measures. This raises the question to which extent inflammatory circuits in MPN depend on activation of innate immune sensors. Here, we investigate whether NLRP3, which precipitates inflammasome assembly upon detection of cellular stress, drives murine JAK2V617F mutant MPN. Deletion of *Nlrp3* within the hematopoietic compartment completely prevents increased IL-1β and IL-18 release in MPN. NLRP3 in JAK2V617F hematopoietic cells, but not in JAK2 wild type radioresistant cells, promotes excessive platelet production via stimulation of the direct thrombopoiesis differentiation pathway, as well as granulocytosis. It also promotes expansion of the hematopoietic stem and progenitor cell compartment despite inducing pyroptosis at the same time. Importantly, NLRP3 inflammasome activation enhances bone marrow fibrosis and splenomegaly. Pharmacological blockade of NLRP3 in fully established disease leads to regression of thrombocytosis, splenomegaly and bone marrow fibrosis. These findings suggest that NLRP3 is critical for MPN development and its inhibition represents a new therapeutic intervention for MPN patients.

The Philadelphia chromosome-negative, classical myeloproliferative neoplasms (for simplicity referred to as MPNs) are caused by somatic mutations in hematopoietic stem cells that promote clonal expansion of myeloid progenitor cells. MPNs encompass several clinicopathologic entities, i.e. polycythemia vera (PV), essential thrombocythemia (ET), and primary myelofibrosis (PMF). The diseases are characterized by myeloid cell proliferation, high risk of thromboembolism, constitutional symptoms and have the potential to progress to acute myeloid leukemia.

The gain-of-function V617F mutation in Janus tyrosine kinase 2 (JAK2) is the most common inducer of MPNs and is detected in 95%

of PV and 50-60% of ET and PMF patients[1]. Notably, JAK2V617F has also been found in 3% of healthy individuals in a Danish general population study[2]. JAK2V617F causes constitutive STAT as well as PI3K/Akt and Ras/Raf/MAPK/ERK signaling. The oncogenic property of JAK2V617F resides in its ability to initiate ligand-independent signaling downstream of the erythropoietin receptor (EPOR), thrombopoietin receptor (TPOR), and granulocyte-stimulating factor receptor (G-CSFR), resulting in erythrocytosis, thrombocytosis, and neutrophilia, respectively[3]. In addition, JAK2 mediates signals from a multitude of other surface receptors, including receptors for chemokines, interleukins, interferons and receptor

tyrosine kinases, many of which are critically involved in inflammatory responses[4].

It is now well recognized that chronic inflammation impacts clinical outcome in MPN. TNFα has been shown to promote clonal dominance of JAK2V617F expressing cells by inhibiting healthy hematopoiesis[5]. Overexpression of IL-8 has been linked to constitutional symptoms, and that of HGF, CXCL9, and IL-1RA to splenomegaly[6]. PV and ET patients have higher serum concentrations of C-reactive protein than healthy individuals, which are associated with an increased incidence of cardiovascular events[7]. Most importantly, IL-8, IL-2R, IL-12, IL-15, and CXCL10 are independently associated with poorer overall survival in PMF[6]. Thus, inflammation in MPN is closely related to clonal selection, symptom burden, thromboembolism and survival.

Recent work has established IL-1β as a major regulator of inflammation in the aging hematopoietic stem cell niche[8] and in MPN[9,10]. Deletion of *Il1b* in a JAK2V617F MPN mouse model reduced serum levels of a broad range of inflammatory cytokines[10]. Importantly, genetic or pharmacological inhibition of IL-1β or its receptor IL-1R1 was sufficient to ameliorate hallmark symptoms of MPN such as splenomegaly and myelofibrosis[9,10]. However, IL-1β is produced by cells as a cytosolic pro-form that depends on cleavage by the proinflammatory protease caspase-1 for its activity and extracellular release. Caspase-1 also matures IL-18 and gasdermin D, which can induce pyroptosis, a form of lytic cell death[11]. In addition it can degrade the transcription factor GATA1, which specifies erythroid cells and megakaryocytes[12]. Activation of caspase-1 requires the assembly of cytoplasmic multiprotein complexes termed inflammasomes[13]. The type of inflammasome driving MPN development remains to be identified.

Inflammasomes consists of a sensor, the adaptor molecule Apoptosis-associated speck-like protein containing a caspase recruitment domain (ASC) and the effector caspase-1. The best characterized inflammasome sensors are NLRP1, NLRP3, NLRC4, and AIM2. NLRP3 (NOD-, LRR-, and pyrin domain-containing 3) is essential for the immune response to numerous pathogens, but is also involved in chronic sterile inflammatory diseases, such as cryopyrin-associated periodic syndrome, type II diabetes, Alzheimer's disease, atherosclerosis, and gout[14]. Recently, it has been implicated in the pathogenesis of myelodysplastic syndrome (MDS)[15] and KRAS mutant myeloid neoplasms[16]. NLRP3 expression is increased in bone marrow cells from MPN patients and NLRP3 inflammasome activation is enhanced in response to various stimuli in monocytes from MPN patients, human JAK2V617F mutant iPSC-derived and murine bone marrow-derived macrophages (BMDM)[17–19].

Here, we define the specific contributions of the inflammasome sensor NLRP3 to the pathogenesis of MPN by adopting two strategies, genetic deletion and pharmacological inhibition, to inactivate NLRP3 in a conditional JAK2V617F-driven mouse model of MPN.

## Results

### MPN patients exhibit increased spontaneous inflammasome activation

We performed cytokine profiling on serum samples from 173 MPN patients from the German Study Group MPN (GSG-MPN) bioregistry and 37 healthy controls. K-means clustering, an unsupervised algorithm, revealed a cluster of 17 cytokines (cluster A) that was upregulated in 43% of MPN patients but only in 13% of healthy controls (Fig. 1a). Notably, this cluster contained IL-1β and IL-18, whose maturation depends on caspase-1, suggesting inflammasome activation. The cluster also included IL-6, whose expression is induced by IL-1β, and TNFα, a priming factor for the NLRP3 inflammasome. We confirmed that IL-1β, IL-18 and TNFα levels were higher in MPN patients than in healthy controls when considered individually (Fig. 1b). In addition, serum concentrations of HMGB1, an alarmin that depends on

inflammasome activation for its extracellular release, was also increased in MPN patients compared to healthy individuals (Fig. 1c). We then tested whether this heightened inflammasome activity in MPN patients is due to increased transcriptional priming. *IL1B* and *CASP1* mRNA was upregulated in PBMCs from MPN patients compared to those from healthy controls (Fig. 1d). To determine whether JAK2V617F promotes transcriptional priming in a cell-intrinsic fashion, we used a published scRNA-seq dataset of hematopoietic stem and progenitor cells (HSPCs) from MPN patients containing single cell genotype information[20]. JAK2V617F mutant HSPC showed stronger expression of an inflammasome gene signature than JAK2 wild type (WT) HSPCs from the same patients (Fig. 1e). These data support the concept that JAK2V617F cell-intrinsically upregulates inflammasome pathway components.

Many of the PV and MF patients in this study were treated with the JAK inhibitor ruxolitinib, which exerts anti-inflammatory in addition to antimyeloproliferative effects. Data from a phase 1-2 trial demonstrate that ruxolitinib influences some serum cytokine levels within 28 days of treatment[21]. Notably, there was no difference in serum IL-1β and IL-18 concentrations between untreated PV and MF patients and those receiving ruxolitinib under real-world conditions in our data set (Fig. 1f).

Altogether, these findings demonstrate spontaneous inflammasome priming and activation in MPN patients.

### ASC speck formation in MPN patients is mediated by NLRP3

A hallmark of inflammasome activation is the ASC speck. We therefore quantified speck formation in peripheral blood mononuclear cells (PBMC) from MPN patients and healthy volunteers by flow cytometry[22]. The percentage of ASC speck positive cells in PBMCs without prior stimulation or after LPS plus nigericin was greater in MPN patients than in healthy individuals (Fig. 2a, b) and positively correlated with serum IL-1β concentrations (Fig. 2c). The increase in ASC speck positive cells was not due to a higher percentage of myeloid cells in MPN patients, as it was present in lymphocytes and myeloid cells (Fig. 2d). To assess the role of NLRP3 in ASC speck formation, we used flow cytometry to examine co-localization of NLRP3 and ASC specks in PBMCs from MPN patients. NLRP3 largely co-stained with ASC specks, indicating the formation of NLRP3-ASC complexes (Fig. 2e, f). To further confirm NLRP3's involvement, we treated MPN patient PBMCs with the NLRP3 inhibitor MCC950. This treatment reduced the proportion of cells with ASC specks (Fig. 2g).

Thus, NLRP3 contributes at least in part to inflammasome activation observed in MPN patients.

### Caspase-1 dependent cytokine secretion in *Jak2^VF* mice depends on the NLRP3 inflammasome

In the *Vav-Cre;Jak2^V617F/+* mouse model (*Jak2^VF* hereafter) of MPN, expression of JAK2V617F induces a PV-like phenotype[23]. We investigated whether inflammasome activation, as observed in MPN patients, also occurs in *Jak2^VF* mice. IL-1β, IL-18, TNFα and HMGB1 serum levels were markedly elevated in *Jak2^VF* relative to WT mice (Fig. 3a). Western blot analysis revealed higher amounts of pro-IL-1β in *Jak2^VF* than WT BMDM even without any stimulation and cleaved IL-1β was reliably detectable in the supernatant of LPS plus nigericin stimulated BMDM from *Jak2^VF* mice but barely evident in those from WT mice (Fig. 3b). To determine whether the expression of NLRP3 itself is dependent on JAK2V617F, we performed immunohistochemistry for NLRP3 on bone marrow sections from *Jak2^VF*, WT and *Nlrp3^−/−* mice. This analysis revealed a robust elevation of NLRP3 staining in *Jak2^VF* mice, supporting the hypothesis that JAK2V617F drives NLRP3 protein expression (Fig. 3d, e). To specifically test the role of the NLRP3 inflammasome in MPN we generated global NLRP3-deficient *Jak2^VF* mice (*Jak2^VF;Nlrp3^−/−*). Strikingly, loss of NLRP3 in *Jak2^VF* mice normalized IL-1β, IL-18 and TNFα serum concentrations (Fig. 3a). Using the

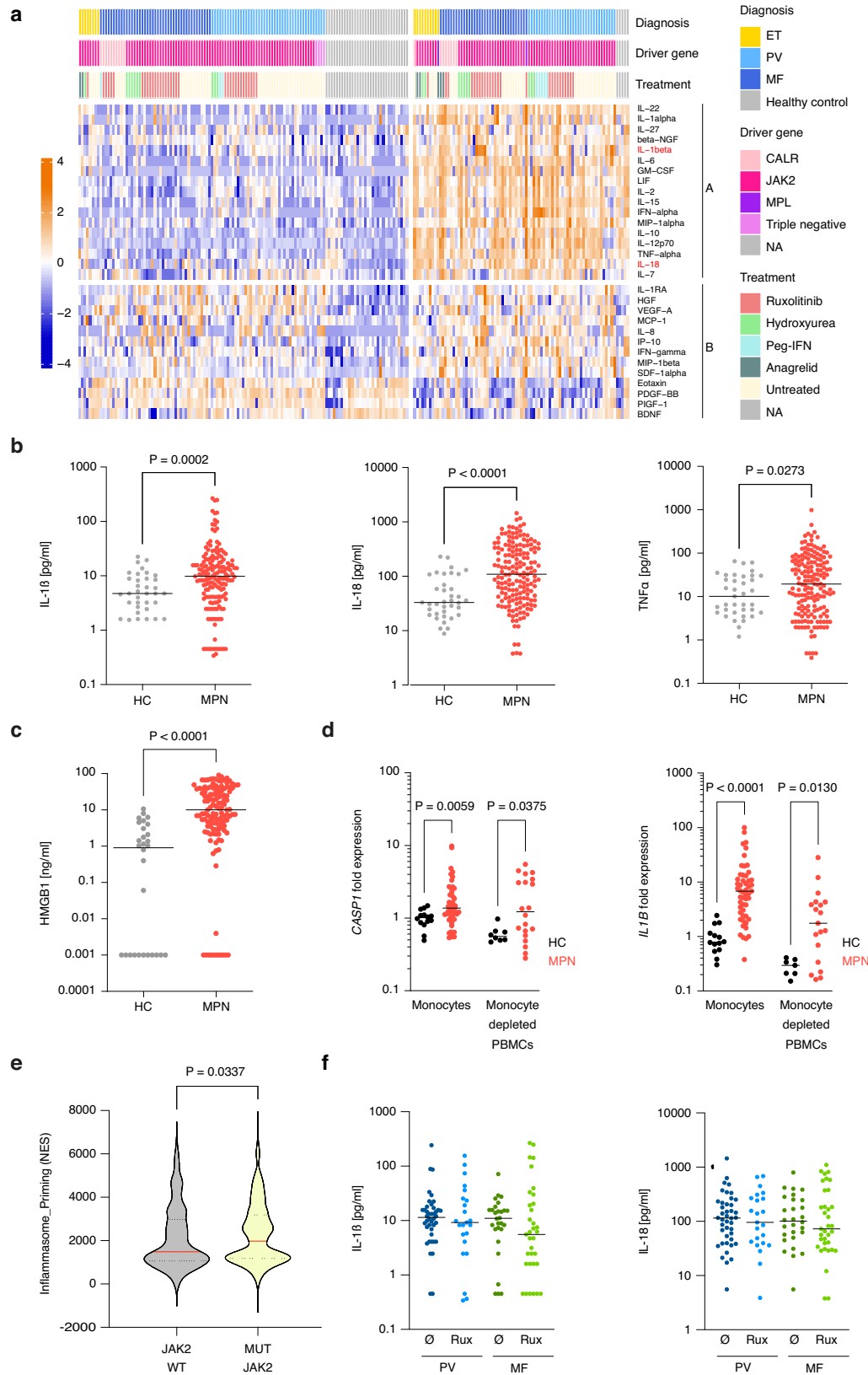

murine myeloid cell line 32D, we found that caspase 1 activation is not solely induced by JAK2V617F, but also by other known MPN driver mutations, including CALRdel52 and CALRins5 (Fig. 3c).

In summary, in a JAK2V617F mutant mouse model of MPN, secretion of caspase-1 dependent cytokines relied on the NLRP3 inflammasome.

## NLRP3 drives thrombocytosis and granulocytosis in murine MPN

$Jak2^{VF}$ mice develop elevated erythrocyte, leukocyte and platelet counts[23]. We sought to investigate the impact of the NLRP3 inflammasome on blood cell production. $Jak2^{VF}$, $Jak2^{VF};Nlrp3^{-/-}$ or WT bone marrow was transplanted into lethally irradiated WT mice (called $Jak2^{VF}$

**Fig. 1 | MPN patients exhibit increased spontaneous inflammasome activation. a** Serum cytokines in healthy controls (n = 37) and MPN patients (n = 173) were measured by Luminex assay. Row- and column-wise K-means clustering with K = 2. Cytokine expression values are Z score standardized. Each column represents one individual. **b** IL-1β, IL-18 and TNFα serum concentrations in healthy controls (HC, n = 37) and MPN patients (n = 173). **c** HMGB1 serum concentration in healthy controls (HC, n = 30) and MPN patients (n = 129). **d** *CASP1* and *IL1B* transcripts were quantified in monocytes from MPN patients (n = 52, including 20 PV, 8 ET and 24 PMF cases) and healthy controls (n = 14); and in monocyte-depleted PBMCs from MPN patients (n = 20, including 4 PV, 5 ET and 11 PMF cases for *CASP1*; n = 19, including 4 PV, 5 ET, 10 PMF for *IL1B*) and healthy controls (n = 8 for *CASP1*; n = 7 for

*IL1B*) by qRT-PCR. Data represent normalized expression values relative to monocytes from healthy controls. **e** Enrichment scores for a gene set comprising *IL1B, IL18, GSDMD, PYCARD* and *CASP1* (Inflammasome_Priming) in JAK2 WT and JAK2V617F mutant single HSPCs isolated from JAK2V617F positive MF patients (n = 8; 4 PMF, 3 post-PV MF, 1 post-ET MF) (GSE122198). Violin plot displays median and quartiles. **f** IL-1β and IL-18 serum concentration in ruxolitinib (Rux) treated and untreated (∅) PV and MF patients. PV (∅ n = 42, Rux n = 23), MF (∅ n = 28, Rux n = 34). Scatter plots in (**b−d, f**) show the median with dots representing single individuals. Statistically significant differences were determined by two-tailed unpaired Mann-Whitney U test. Source data are provided as a Source Data file.

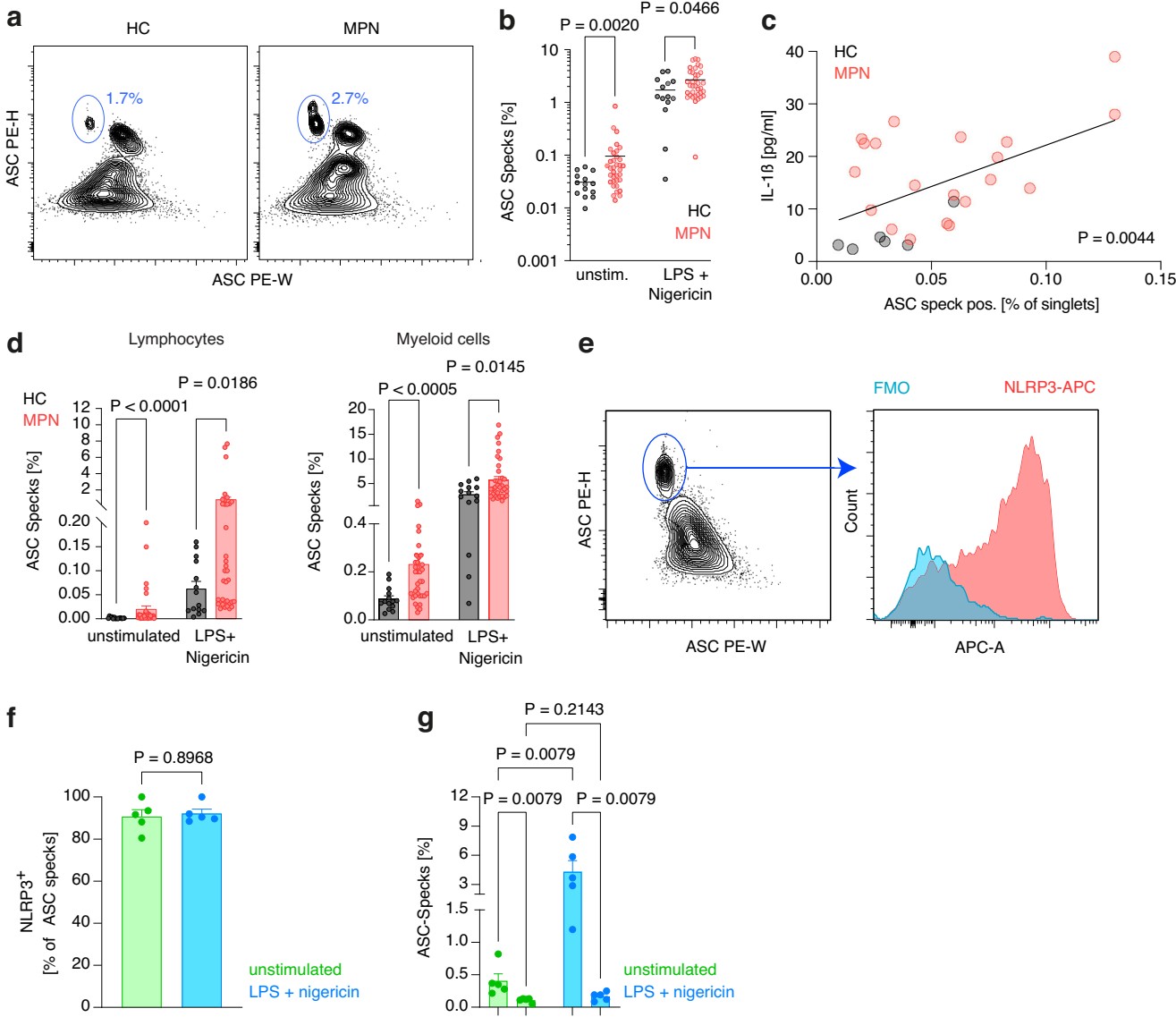

**Fig. 2 | ASC speck formation in MPN patients is mediated by NLRP3.**
**a** Representative ASC staining profiles and gating of ASC speck positive PBMCs in a healthy control (HC) and a MPN patient after LPS plus nigericin stimulation. Values indicate percentage of PBMCs (mean). **b** Percentage of ASC speck positive cells in PBMCs from healthy controls (HC, n = 14) or MPN patients under unstimulated conditions (n = 35; PV 18, ET 7, PMF 10) or following LPS plus nigericin stimulation (n = 36; PV 19, ET 7, PMF 10). Dots represent single individuals and horizontal lines the mean. **c** Correlation of serum IL-1β concentration and percentage of ASC speck positive cells in PBMCs from healthy controls (HC, n = 6) and MPN patients (n = 20). Dots represent single individuals. **d** Percentage of lymphocytes and myeloid cells with aggregated ASC in PBMCs from healthy controls (HC, n = 14) or MPN patients (n = 36) under unstimulated conditions or following LPS plus nigericin stimulation.

**e** Representative example of co-localization of ASC specks and NLRP3 in PBMCs from a MPN patient (FMO, fluorescence minus one control). **f** Percentage of ASC specks that stain positive for NLRP3 in PBMCs from MPN patients (n = 5) under unstimulated conditions or following LPS plus nigericin stimulation. **g** Percentage of ASC speck positive cells in PBMCs from MPN patients (n = 5) under unstimulated conditions or following LPS plus nigericin stimulation, with or without the NLRP3 inhibitor MCC950. Scatter bar plots in (**d, f, g**) show mean + SEM with dots representing single individuals. Statistically significant differences were determined by two-tailed unpaired Mann-Whitney U test (**b, d, f**), two-tailed Pearson correlation (**c**) or Kruskall-Wallis test with two-sided Dunn's multiple comparisons test (**g**). Source data are provided as a Source Data file.

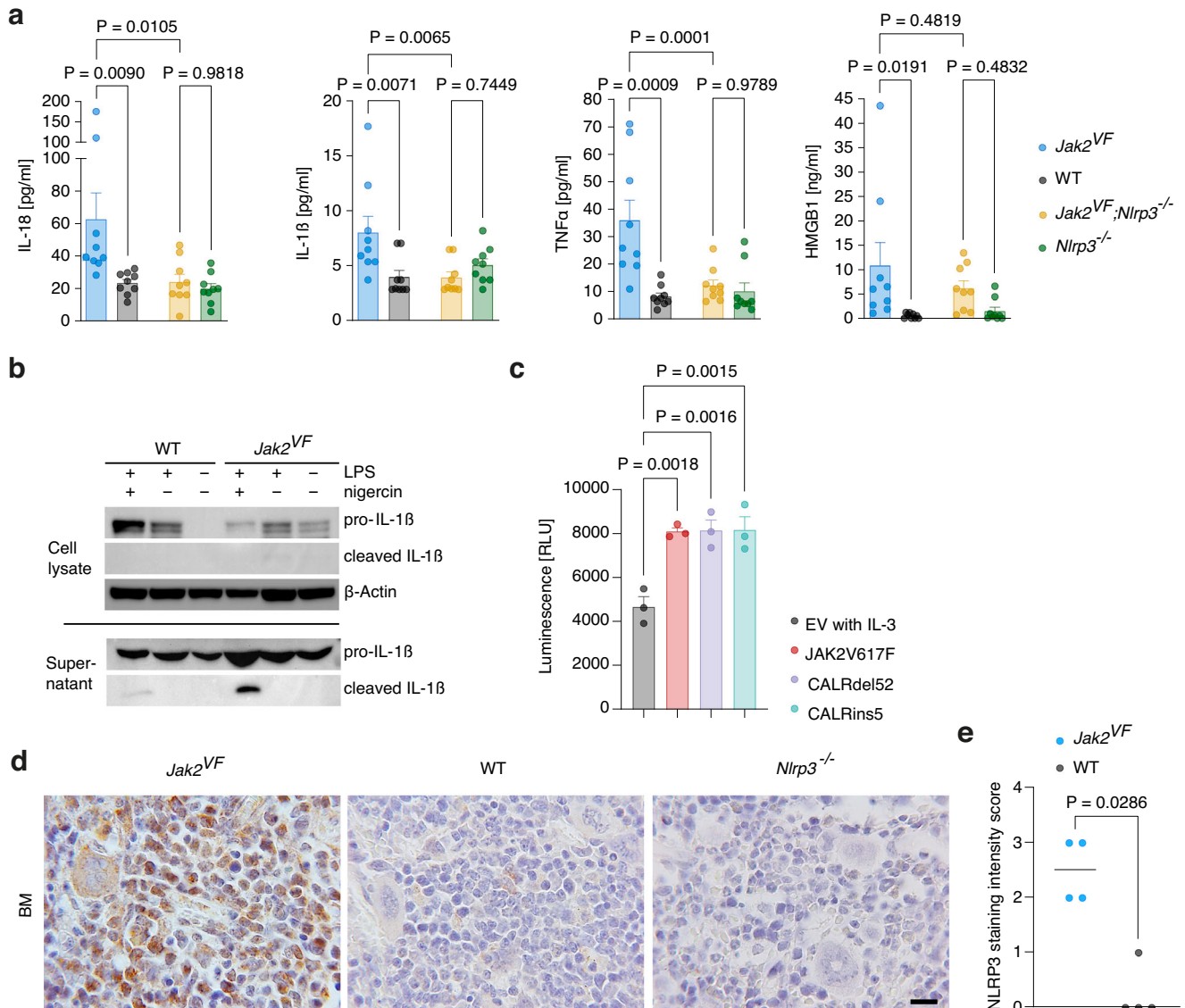

**Fig. 3 | Caspase-1 dependent cytokine secretion in *Jak2^{VF}* mice depends on the NLRP3 inflammasome. a** IL-1β, IL-18, TNFα and HMGB1 serum concentrations in *Jak2^{VF}*, WT, *Jak2^{VF};Nlrp3^{-/-}* and *Nlrp3^{-/-}* mice (n = 9 mice/group). Dots represent individual mice. **b** BMDMs from *Jak2^{VF}* and WT mice were stimulated or left untreated as indicated, and cell lysates and supernatants were assessed for pro-IL-1β and cleaved IL-1β by immunoblotting. The blot is representative of 2 independent experiments. **c** Caspase 1 activation in 32D cells expressing JAK2V617F, CALRdel52, CALRins5 or an empty vector (EV, no oncogene). For 32D EV cells, the medium was supplemented with murine IL-3. Each dot represents an independent experiment using separately cultured cell populations (n = 3 per group). **d** Representative images of bone marrow sections from *Jak2^{VF}*, WT and *Nlrp3^{-/-}* mice at 52 weeks of age stained for NLRP3. Scale bar equals 100 μm. **e** NLRP3 staining intensity score of bone marrow sections from *Jak2^{VF}* and WT mice. Dots represent individual mice and horizontal lines the median (n = 4 mice/group). Scatter bar plots in (**a, c**) show mean + SEM. Statistically significant differences were determined by one-way ANOVA with two-sided Holm-Šidák multiple comparison test (**a**) or one-way ANOVA with two-sided Dunnett's multiple comparison test (**c**), or by two-tailed unpaired Mann-Whitney U test (**e**). Source data are provided as a Source Data file.

BM, *Jak2^{VF};Nlrp3^{-/-}* BM and WT BM mice hereafter) and blood was obtained by the submandibular vein method repeatedly over 25 weeks. Platelet numbers were greatly reduced in *Jak2^{VF};Nlrp3^{-/-}* BM compared to *Jak2^{VF}* BM mice; in fact, they were indistinguishable from those in WT BM mice (Fig. 4a). Hemoglobin and, for the most part, leukocyte levels were not affected by the deletion of *Nlrp3*. However, leukocyte counts are known to depend on the method of blood acquisition, with cardiac and vena cava bleeds providing the most consistent results[24]. Therefore, after 26 weeks, mice were sacrificed and blood from the abdominal vena cava was collected. In this analysis, platelets and leukocytes were both lower in *Jak2^{VF};Nlrp3^{-/-}* BM than in *Jak2^{VF}* BM mice (Fig. 4b). The drop in leukocytes was mainly due to a decrease in neutrophils, basophils, and eosinophils. Platelet leukocyte aggregates have been associated with thrombosis in MPN and are correlated with

platelet counts[25]. As expected, neutrophil-platelet and inflammatory monocyte-platelet aggregates were decreased in *Jak2^{VF};Nlrp3^{-/-}* BM relative to *Jak2^{VF}* BM mice (Fig. 4c).

A recent report suggested that NLRP3 may promote hematopoietic stem cell engraftment after transplantation[26]. In this study, platelets and leukocytes were reduced in mice transplanted with *Nlrp3^{-/-}* bone marrow compared to those receiving WT bone marrow until 3 weeks after transplantation, but the difference disappeared after 4 weeks. Thus, an engraftment defect due to NLRP3 deficiency was an unlikely explanation for the lower platelet and leukocyte counts in *Jak2^{VF};Nlrp3^{-/-}* BM than in *Jak2^{VF}* BM mice, which persisted at least until 26 weeks after transplantation. Nevertheless, to investigate this possibility, we analyzed unmanipulated *Jak2^{VF}* and *Jak2^{VF};Nlrp3^{-/-}* mice (i.e. non-transplanted). *Jak2^{VF};Nlrp3^{-/-}*

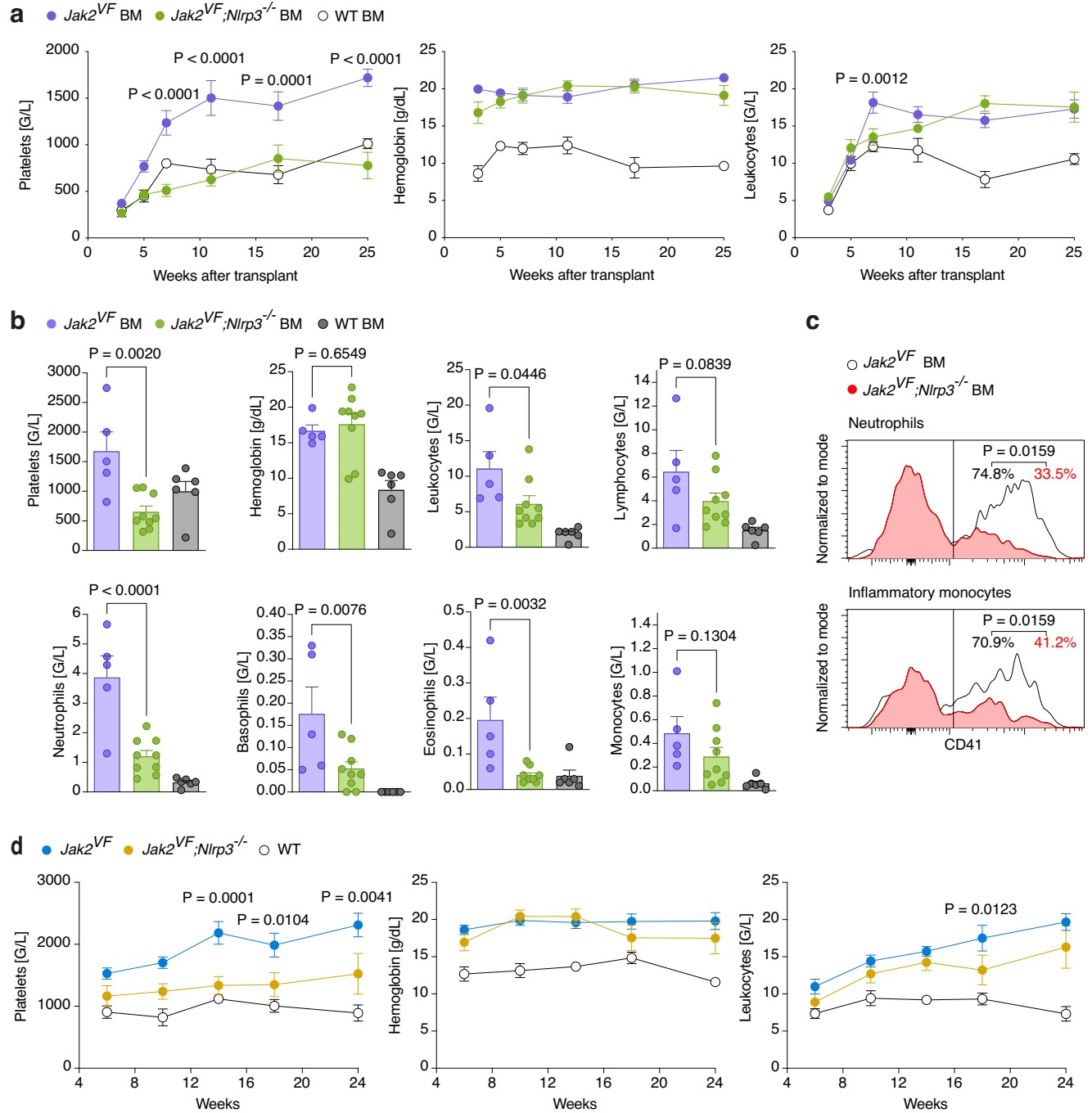

**Fig. 4 | NLRP3 drives thrombocytosis and granulocytosis in murine MPN.**
**a** Blood counts of *Jak2*[VF] BM (n = 12), *Jak2*[VF];*Nlrp3*[-/-] BM (n = 10) and WT BM (n = 6) mice. Blood was drawn by submandibular method. For clarity, only the significant differences between *Jak2*[VF] BM and *Jak2*[VF];*Nlrp3*[-/-] BM are shown. **b** Differential blood count of *Jak2*[VF] BM (n = 5), *Jak2*[VF];*Nlrp3*[-/-] BM (n = 9) and WT BM (n = 6) mice 26 weeks after bone marrow transplantation. Blood was collected from the vena cava. **c** Staining histograms display gating of leukocyte-platelet aggregates in *Jak2*[VF] BM (n = 5) and *Jak2*[VF];*Nlrp3*[-/-] BM (n = 4) mice. Values indicate the mean percentage of neutrophil-platelet and inflammatory monocyte-platelet aggregates. **d** Blood counts of non-transplanted *Jak2*[VF] (n = 8), *Jak2*[VF];*Nlrp3*[-/-] (n = 10) and WT (n = 9) mice. Blood was drawn by submandibular method. The significant differences between *Jak2*[VF] and *Jak2*[VF];*Nlrp3*[-/-] are displayed. Plots show mean + SEM. In scatter plots, each dot represents an individual mouse. Statistically significant differences were determined by Mixed-effects model with two-sided Dunnett's multiple comparisons test (**a**, **d**), one-way ANOVA with two-sided Holm-Šidák multiple comparison test (**b**) and two-tailed unpaired Mann-Whitney U test (**c**). Source data are provided as a Source Data file.

mice showed lower platelets and granulocytes than *Jak2*[VF] mice, consistent with the results obtained in transplanted animals (Figs. 4d and S1a).

To investigate whether NLRP3 in non-hematopoietic radioresistant cells (such as epithelial and stromal cells) affects hematopoiesis in MPN we transplanted lethally irradiated WT and *Nlrp3*[-/-] mice with *Jak2*[VF] bone marrow. Lack of NLRP3 in radioresistant cells caused a slight reduction in platelets and leukocytes at week 5 after transplantation but had no effect beyond the engraftment period (Fig. S1b).

We conclude that *Nlrp3* deletion in hematopoietic cells impedes thrombocytosis and granulocytosis in JAK2V617F-induced murine MPN.

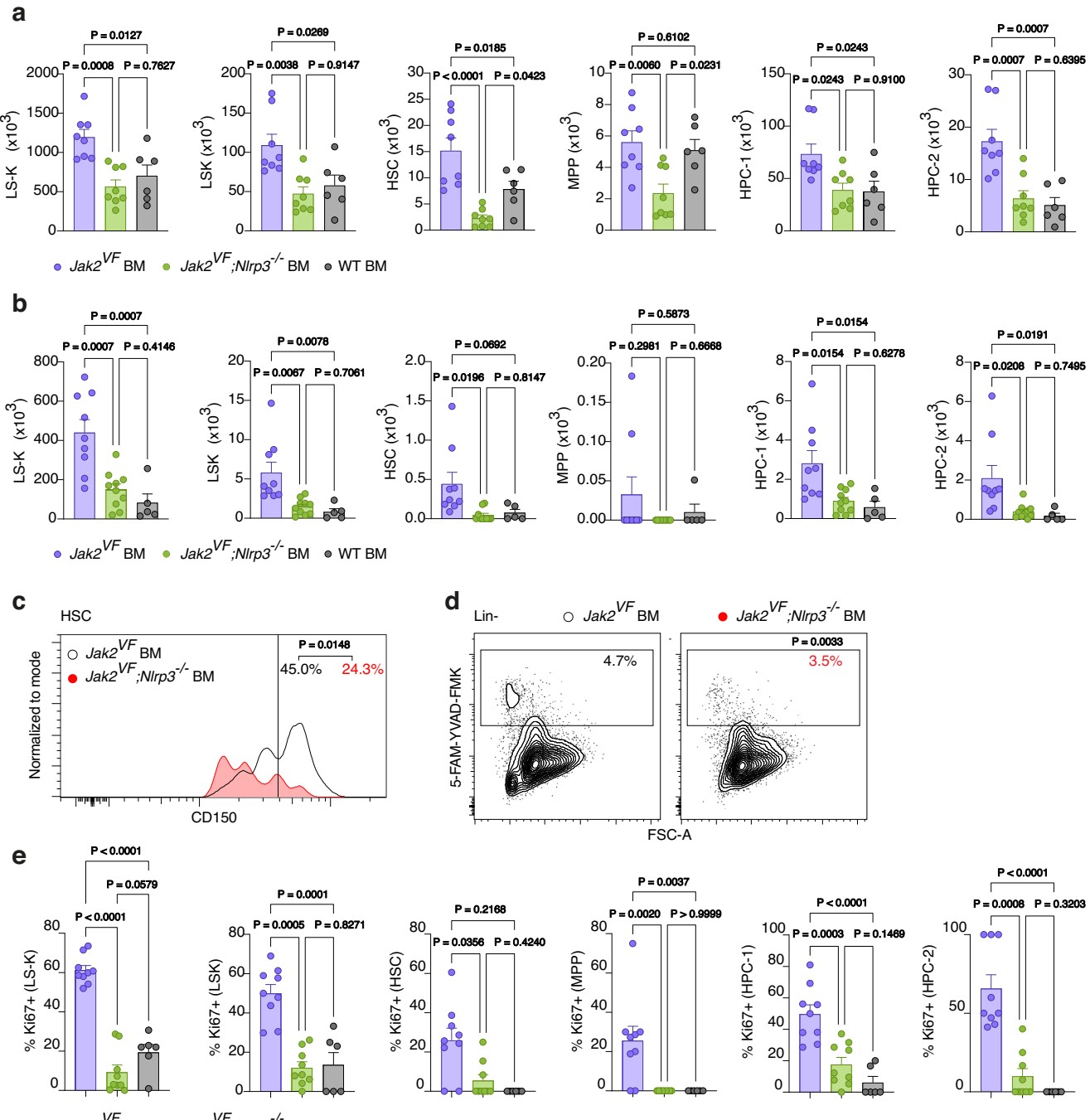

**Fig. 5 | The NLRP3 inflammasome promotes HSPC expansion and myeloid differentiation. a** HSPC subset counts in the bone marrow from both femurs of *Jak2*VF BM (n = 8), *Jak2*VF;*Nlrp3*-/- BM (n = 8) and WT BM (n = 6) mice. **b** As in (A) but in spleens of *Jak2*VF BM (n = 9), *Jak2*VF;*Nlrp3*-/- BM (n = 10) and WT BM (n = 5) mice. **c** Representative CD150 staining histograms of bone marrow HSCs in *Jak2*VF BM (n = 8) and *Jak2*VF;*Nlrp3*-/- BM (n = 8) mice. Values indicate the mean percentage of HSCs with high CD150 expression, indicating a myeloid bias. **d** Contour plots display gating of pyroptotic bone marrow HSPCs (Lin − ) of *Jak2*VF BM (n = 8) and

*Jak2*VF;*Nlrp3*-/- BM (n = 8) mice. Values indicate the mean percentage of HSPCs that undergo pyroptosis. **e** Percentage of Ki67 positive cells in HSPC subsets in the bone marrow of *Jak2*VF BM (n = 9), *Jak2*VF;*Nlrp3*-/- BM (n = 9) and WT BM (n = 6) mice. Scatter bar plots in (**a**, **b**, **e**) show mean + SEM with dots representing individual mice. Statistically significant differences were determined by one-way ANOVA with two-sided Holm-Šidák multiple comparison test (**a**, **b**, **e**) and two-tailed unpaired Mann-Whitney U test (**c** and **d**). Source data are provided as a Source Data file.

## The NLRP3 inflammasome promotes HSPC expansion and myeloid differentiation

JAK2V617F leads to an expansion of the HSPC compartment in MPN. The size of the LS-K cell population in the bone marrow was diminished in *Jak2*VF;*Nlrp3*-/- BM relative to *Jak2*VF BM mice and comparable to that in WT BM mice (Fig. 5a and gating schematic in Fig. S2). Likewise, there were fewer LSK cells in the bone marrow of

*Jak2*VF;*Nlrp3*-/- BM than of *Jak2*VF BM mice. All LSK subsets, i.e. HSCs (hematopoietic stem cells), MPPs (multipotent progenitor) and HPCs (hematopoietic progenitor cells), in *Jak2*VF;*Nlrp3*-/- BM mice were reduced to numbers similar to or lower than those observed in WT BM mice. We made largely similar findings in the spleen (Fig. 5b). Thus, the NLRP3 inflammasome drives enlargement of the HSPC pool in the bone marrow and spleen in murine MPN.

HSCs represent a functionally heterogenous population. In MPN, HSCs show myeloid-biased differentiation. The expression level of CD150 on HSCs can be used to distinguish myeloid- from lymphoid-biased HSCs[27]. Notably, deficiency for NLRP3 decreased the frequency of HSCs expressing high levels of CD150 in murine MPN (Fig. 5c), suggesting that NLRP3 imparts a myeloid skew, in keeping with published data on IL-1β[28].

Pyroptosis is an inflammatory form of cell death caused by caspase-1 cleavage of gasdermin D following inflammasome activation. We asked whether NLRP3 induces pyroptosis in HSPCs. To identify pyroptotic cells, we used the fluorescently labeled caspase-1/-4/-5 inhibitor 5-FAM-YVAD-FMK and scatter characteristics. Surprisingly, NLRP3 in JAKV617F mutant MPN indeed stimulated pyroptosis in HSPCs (Fig. 5d) even though it increased the HSPC population size. To resolve this discrepancy, we examined cell proliferation by Ki67 staining and found that NLRP3 enhanced proliferation across all HSPC subsets (Fig. 5e). We attempted to directly detect pyroptosis by staining for cleaved gasdermin D but encountered high levels of nonspecific staining with all tested antibody clones.

To address cell-type specific inflammasome activation, we performed intracellular pro-IL-1β staining in both mature and immature bone marrow cells from *Jak2^VF* and *Vav-Cre* mice. Among mature cell populations, we observed a considerable increase in pro-IL-1β production in B cells when JAK2V617F was present (Fig. S3a). This indicates that B cells may contribute to the NLRP3-driven inflammatory response in MPN. Interestingly, JAK2V617F-positive neutrophils did not display higher levels of pro-IL-1β compared to their *Vav-Cre* counterparts, arguing against the notion that neutrophils are key drivers of NLRP3-dependent disease progression in this model. Among immature cell populations, JAK2V617F elevated pro-IL-1β levels across all examined populations, with statistical significance reached in CMPs (common myeloid progenitors), MEPs (megakaryocyte-erythroid progenitors), and MkPs (megakaryocyte progenitors) (Fig. S3a). Staining with 5-FAM-YVAD-FMK confirmed higher caspase activity in JAK2V617F mutant CMPs, MEPs, and MkPs compared to *Vav-Cre* cells (Fig. S3b). These findings suggest that a broad range of myeloid and lymphoid cell populations, including B cells and several progenitor populations, contribute to the NLRP3-mediated inflammatory cascade in the context of JAK2V617F-induced MPN.

Overall, NLRP3 promoted expansion and myeloid skewing of the HSPC pool in murine JAK2V617F-driven MPN.

## NLRP3 stimulates the direct thrombopoiesis pathway

NLRP3 deficiency potently suppressed platelet production in JAK2V617F-induced MPN. To better understand how NLRP3 regulates thrombopoiesis, we quantified megakaryocytes and MkPs in the bone marrow and spleen. Megakaryocytes were manually counted on tissue sections, as they tend to be underrepresented in flow cytometric assays because of their large cell size and fragility. *Jak2^VF;Nlrp3^-/-* BM mice contained substantially fewer megakaryocytes and MkPs than *Jak2^VF* BM mice in bone marrow and spleen, respectively (Fig. 6a–c). Deletion of *Nlrp3* in *Jak2^VF* mice also reduced megakaryocyte size and dysplastic features of the nuclei (Fig. 6d). Importantly, thrombopoietin serum concentrations were similar in *Jak2^VF* BM and *Jak2^VF;Nlrp3^-/-* BM mice (Fig. 6e). Thus, NLRP3 deficiency did not lower platelet, megakaryocyte and MkP counts by reducing thrombopoietin levels.

Until recently, the thrombopoiesis differentiation pathway was thought to proceed from HSCs via CMPs to MkPs. However, several studies have now provided evidence that megakaryocytes can also be replenished by direct differentiation of HSCs into MkPs[29–31]. Both the long and the short route are considered to be active simultaneously[30]. CD48 expression on MkPs distinguishes MkPs directly connected to HSCs (CD48^lo MkPs) from those that differentiate from CMPs (CD48^hi MkPs) (Fig. 6f). The frequency of MkPs that were CD48^lo was decreased in *Jak2^VF;Nlrp3^-/-* BM compared with *Jak2^VF* BM mice (Fig. 6g), indicating

that loss of NLRP3 impairs platelet production via the short route. Remarkably, deficiency for NLRP3 selectively reduced the frequency of actively cycling CD48^lo but not CD48^hi MkPs, as assessed by Ki67 staining (Fig. 6h). To elucidate the mechanism by which NLRP3 influences MkP development, we injected WT mice with IL-1β and monitored CD48^lo and CD48^hi MkP subsets. We found that IL-1β increased the frequency of CD48^lo MkPs while decreasing CD48^hi MkPs within the bone marrow MkP population (Fig. 6i). These findings suggest that NLRP3 promotes MkP production via the short route through IL-1β.

We also examined how NLRP3 deficiency alters gene expression in HSCs and MkPs by 3′-RNA sequencing of sorted cells from *Jak2^VF* and *Jak2^VF;Nlrp3^-/-* mice. The analysis revealed that deleting *Nlrp3* enhances the expression of an oxidative phosphorylation (OXPHOS) signature (Fig. S4a-h). We validated this metabolic shift using a biochemical assay measuring OXPHOS and glycolysis. Inhibiting OXPHOS with oligomycin confirmed that mitochondrial ATP production was higher in *Jak2^VF;Nlrp3^-/-* than in *Jak2^VF* bone marrow cells (Fig. S4i), consistent with reports showing that NLRP3 promotes glycolysis via IL-1β[32].

Collectively, these data demonstrate that NLRP3 stimulates the direct thrombopoiesis pathway in murine MPN, at least in part by inducing proliferation of associated CD48^lo MkPs.

## NLRP3 inflammasome activation propagates bone marrow fibrosis and splenomegaly

*Jak2^VF* mice develop bone marrow fibrosis and, as a consequence of extramedullary hematopoiesis, splenomegaly. It remained unclear how NLRP3 would affect these major clinical manifestations of MPN. Because bone marrow transplantation delays the onset of fibrosis we assessed non-transplanted *Jak2^VF*, *Jak2^VF;Nlrp3^-/-* and WT mice. Scoring of silver-stained tissue sections revealed that fibrosis development in the bone marrow and spleens of *Jak2^VF* mice was greatly ameliorated in the absence of NLRP3 (Fig. 7a, b). Expression of the fibrosis-related gene *COL1A1*, which encodes for type I collagen, was also decreased by deleting *Nlrp3* (Fig. 7c). Consistent with this, splenomegaly was reduced in *Jak2^VF;Nlrp3^-/-* mice compared to *Jak2^VF* mice (Fig. 7d, e). The attenuation of fibrosis caused by the lack of NLRP3 aligned with the lower number of megakaryocytes in *Jak2^VF;Nlrp3^-/-* mice, which are key drivers of bone marrow fibrosis[33].

## Pharmacological blockade of NLRP3 alleviates thrombocytosis, bone marrow fibrosis and splenomegaly

In the *Nlrp3* gene deletion studies, *Jak2^VF* hematopoietic cells were already NLRP3-deficient in utero. We asked whether blocking NLRP3 after disease is already established would also lessen myeloproliferation. For this purpose, we used the novel oral NLRP3 inhibitor IFM-2384. *Jak2^VF* BM mice received a chow formulation of IFM-2384 or control chow for 20 weeks starting 16 weeks after transplantation. On target efficacy was proven by reduced serum levels of IL-1β, TNFα and HMGB-1 in NLRP3 inhibitor-exposed *Jak2^VF* BM mice (Fig. 8a). When blood was drawn from the submandibular vein, we found that in the IFM-2384 group, platelets decreased continuously from 914 G/L before to 333 G/L after 16 weeks of therapy (Fig. 8b). In contrast, thrombocytosis remained stable in the control group. There was also a trend for lower leukocytes in mice fed with IFM-2384 compared to those with control chow but this effect did not reach statistical significance. After 20 weeks of treatment, the mice were sacrificed and blood was obtained from the abdominal vena cava, which showed that NLRP3 inhibition decreased platelets and leukocytes, including neutrophils (Fig. S5). In keeping with the platelet data, there were fewer bone marrow megakaryocytes in *Jak2^VF* BM mice fed IFM-2384 containing chow compared to those given control chow (Fig. 8d). Furthermore, IFM-2384 administration restricted HSPC expansion in murine MPN (Fig. 8c). Finally, NLRP3 blockade considerably mitigated splenomegaly and bone marrow fibrosis in *Jak2^VF* BM mice (Fig. 8e, f). The

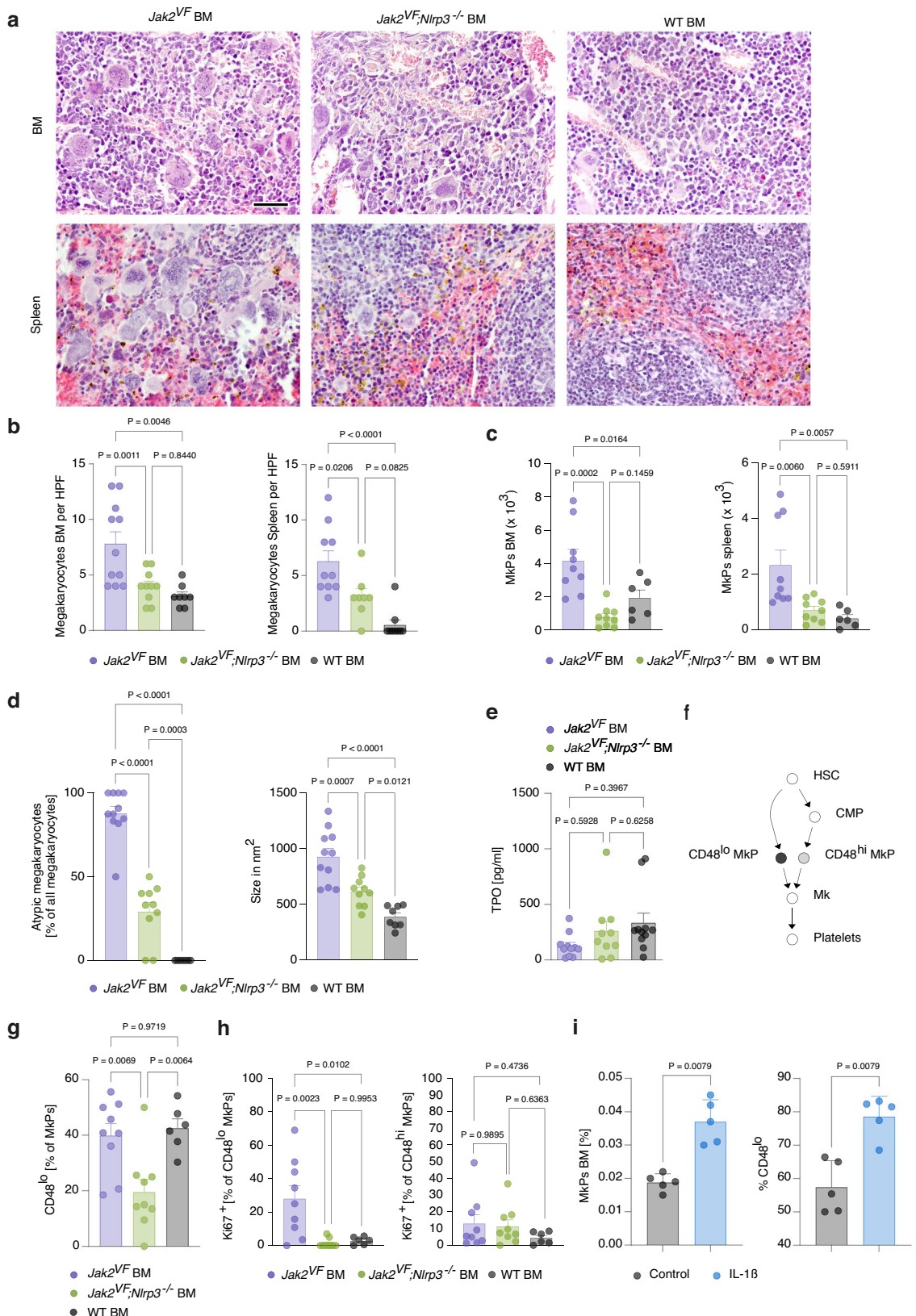

reduction in bone marrow fibrosis observed in *Jak2^VF* BM mice treated with IFM-2384 was likely due to the inhibitor's action on mutant hematopoietic cells, rather than on the stromal compartment. This is supported by transplantation experiments indicating that NLRP3 in radioresistant non-hematopoietic cells does not influence BM fibrosis (Fig. S1c).

These inhibitor studies further support the notion that NLRP3 is a viable therapeutic target in MPN.

## Discussion

Our study provides detailed insights into the essential contributions of the NLRP3 inflammasome to MPN development. By targeting NLRP3

**Fig. 6 | NLRP3 stimulates the direct thrombopoiesis pathway. a** Representative images of H&E-stained bone marrow and spleen sections of *Jak2*$^{VF}$ BM, *Jak2*$^{VF}$;*Nlrp3*$^{-/-}$ BM and WT BM mice at 26 weeks after bone marrow transplantation illustrating the megakaryocyte density. Scale bar equals 50 μm. **b** Megakaryocyte counts in the bone marrow (left) of *Jak2*$^{VF}$ BM (n = 11), *Jak2*$^{VF}$;*Nlrp3*$^{-/-}$ BM (n = 10) and WT BM (n = 8) mice, and in the spleen (right) of *Jak2*$^{VF}$ BM (n = 10), *Jak2*$^{VF}$;*Nlrp3*$^{-/-}$ BM (n = 8) and WT BM (n = 9) mice (HPF, high-power field). **c** MkP counts in *Jak2*$^{VF}$ BM (n = 9), *Jak2*$^{VF}$;*Nlrp3*$^{-/-}$ BM (n = 9) and WT BM (n = 6) mice. **d** Megakaryocyte dysplasia was assessed based on atypical megakaryocyte morphology (left) and size (right) in bone marrow sections from *Jak2*$^{VF}$ BM (n = 11), *Jak2*$^{VF}$;*Nlrp3*$^{-/-}$ BM (n = 10) and WT BM (n = 8) mice. Morphological features of atypical megakaryocytes included irregular cell shape, hypo- or hyperlobulated nuclei, abnormal nuclear-to-cytoplasmic ratio, and clustering. **e** Thrombopoietin (TPO) serum concentration in *Jak2*$^{VF}$ BM (n = 11), *Jak2*$^{VF}$;*Nlrp3*$^{-/-}$ BM (n = 10) and WT BM (n = 11) mice. **f** Schematic of the direct and conventional thrombopoiesis pathway. **g** Percentage of MkPs that are CD48$^{lo}$ in *Jak2*$^{VF}$ BM (n = 9), *Jak2*$^{VF}$;*Nlrp3*$^{-/-}$ BM (n = 9) and WT BM (n = 6) mice. **h** Percentage of proliferating (Ki67$^+$) CD48$^{lo}$ (left) and CD48$^{hi}$ (right) MkPs in *Jak2*$^{VF}$ BM (n = 9), *Jak2*$^{VF}$;*Nlrp3*$^{-/-}$ BM (n = 9) and WT BM (n = 6) mice. **i** Percentage of MkPs among live cells in the bone marrow (left) and percentage of CD48$^{lo}$ cells within the MkP population (right) of WT mice at 8 h after IL-1β injection (20 μg/kg, n = 5) or of untreated controls (n = 5). Scatter bar plots show mean + SEM with dots representing individual mice. Statistically significant differences were determined by one-way ANOVA with two-sided Holm-Šidák multiple comparison test (**b–e, g, h**) or two-tailed unpaired Mann-Whitney U test (**i**). Source data are provided as a Source Data file.

genetically or pharmacologically in a JAK2V617F mutant mouse model, we formally establish that NLRP3 potently promotes malignant thrombocytosis and, to a lesser extent, granulocytosis. Notably, deletion of *Nlrp3* specifically impaired platelet production via the direct thrombopoiesis pathway. The loss of NLRP3 protected *Jak2*$^{VF}$ mice from fibrosis in the bone marrow and spleen, highlighting the importance of inflammasome activation in fibrogenesis. Splenomegaly, another key manifestation of MPN, was also markedly diminished in the absence of NLRP3.

The model of thrombopoiesis has recently been considerably advanced based on the finding that MkPs can arise directly from HSCs without first proceeding through CMPs[29–31]. In a mouse model of mutant *Calr*-driven ET, it was recently shown that megakaryocyte differentiation mainly occurred via a short route that directly links HSCs with MkPs[34]. Furthermore, single-cell transcriptomic data in MPN patients revealed that a direct differentiation route from HSCs to MkPs is aberrantly expanded compared with healthy individuals[35]. These studies provide evidence that the direct thrombopoiesis pathway may be the predominant contributor to increased platelet output in MPN. The drivers that increase platelet production via this trajectory in MPN have not been explored. Our study suggests, based on the differential expression of CD48 on MkPs, that the NLRP3 inflammasome is a major stimulator of the direct thrombopoiesis pathway in MPN.

NLRP3 might promote MPN development through several mechanism, including maturation of IL-1β and IL-18, initiation of pyroptosis by cleaving the pore-forming cell death executor gasdermin D and degradation of the hematopoietic lineage specifying transcription factor GATA1. Recently, the contribution of IL-1β and IL-1R1 were evaluated in MPN[9,10]. In these studies, deficiency for IL-1β or IL-1R1 attenuated bone marrow fibrosis and splenomegaly. These defects in the IL-1 pathway also impaired granulocyte and platelet production. In our work, NLRP3 exerted similar outcomes in JAK2V617F positive MPN, indicating that the phenotype of *Jak2*$^{VF}$ mice lacking NLRP3 is at least partly due to the reduction in mature IL-1β.

Inflammasome activation triggers pyroptosis via cleavage of gasdermin D by caspase 1. In MDS, the NLRP3 inflammasome has been implicated in the pathogenesis of cytopenia because it induced pyroptosis in HSPCs isolated from MDS patients[15]. We show that in MPN, NLRP3 also provokes pyroptosis in HSPCs but this did not cause cytopenia. On the contrary, NLRP3 rather promoted myeloproliferation. Therefore, other effects of NLRP3 appear to be more dominant over the impairment of hematopoiesis by pyroptosis in MPN. In the case of thrombopoiesis, it is noteworthy that IL-1β upregulates the megakaryocytic transcription factors NF-E2 and GATA1, as well as c-Fos and c-Jun, which are components of the dimeric activating protein-1 (AP-1) transcription-factor complex, thereby directly stimulating megakaryopoiesis[36].

The transcription factor GATA1, which enforces megakaryocytic and erythroid differentiation, has been identified as a target of caspase-1[12]. Caspase-1 inhibition upregulates GATA1 protein in mouse HSPCs, thereby promoting erythrocytosis. Degradation of GATA1 downstream of NLRP3 inflammasome activation has been proposed as a mechanism leading to anemia in MDS[37]. However, in our study of MPN in mice, NLRP3 had no effect on blood hemoglobin content and thus on erythrocytosis, although it was clearly activated. Overall, this highlights that the function of NLRP3 in MPN cannot readily be inferred from other diseases and its role has to be tested in vivo for each individual disease.

In addition to NLRP3, other inflammasomes may regulate malignant hematopoiesis in MPN. Expression of AIM2 is induced by JAK2V617F[38] and in a mouse model of *Jak2* mutant clonal hematopoiesis, *Aim2* deficiency reduced atherosclerosis in an IL-1β-dependent manner[18]. Furthermore, NLRP1a has been found to trigger pyroptosis in HSPCs during hematopoietic stress caused by chemotherapy or viral infection leading to prolonged cytopenia[39]. We here focused on NLRP3 because its expression is increased in MPN patients[17]. We show that serum concentrations of IL-1β and IL-18 in *Jak2*$^{VF}$ mice lacking NLRP3 are reduced to those in WT mice. This does not exclude the possibility that other inflammasomes are also involved in the development of MPN, but it does indicate that NLRP3 is an important inflammasome sensor in this disease.

Formation of the NLRP3 inflammasome complex is widely considered to involve a two-signal mechanism, with signal 1 required for transcriptional upregulation of NLRP3 and other inflammasome components (also known as 'priming'). Subsequently, signal 2 can trigger NLRP3 inflammasome activation leading to caspase-1 activation. Signal 1 is typically induced by cytokine receptors or Toll-like receptor ligands, but the stimuli for signal 2 are more diverse and a unifying mechanism has yet to be defined[14]. Notably, numerous studies have now established that two signals are not always needed for assembly of the NLRP3 inflammasome, especially in vivo[40,41]. Our work reveals that JAK2V617 causes inflammasome priming, likely in a cell-intrinsic fashion. This is consistent with the fact that JAK2V617 induces NFκB signaling[42]. Whether a dedicated signal 2 is essential for NLRP3 activation in MPN remains to be elucidated in future studies. It is worth pointing out that in other myeloid malignancies reactive oxygen species (ROS) have been linked to NLRP3 activation[15,16], but ROS inhibitors appear to rather block priming than activation[43].

Our work has also limitations. In our mouse models all hematopoietic cells were of the *Jak2*$^{VF}$ genotype. Thus, it was not possible to assess whether NLRP3 activation promotes clonal outgrowth of JAK2V617 mutant cells. We also did not investigate the effect of the NLRP3 inflammasome on WT hematopoiesis. Because a Western-type calorically rich diet induces Nlrp3-dependent inflammation, we anticipate that such an analysis would need to take into account dietary conditions[44].

In conclusion, we have delineated unique immunopathological effects of the NLRP3 inflammasome in JAK2 mutant MPN. Although more work is needed to define its regulation, our data confirm NLRP3 as a therapeutic target in patients with MPN, particularly those in whom excessive platelet production is of primary concern, such as in ET and prefibrotic PMF.

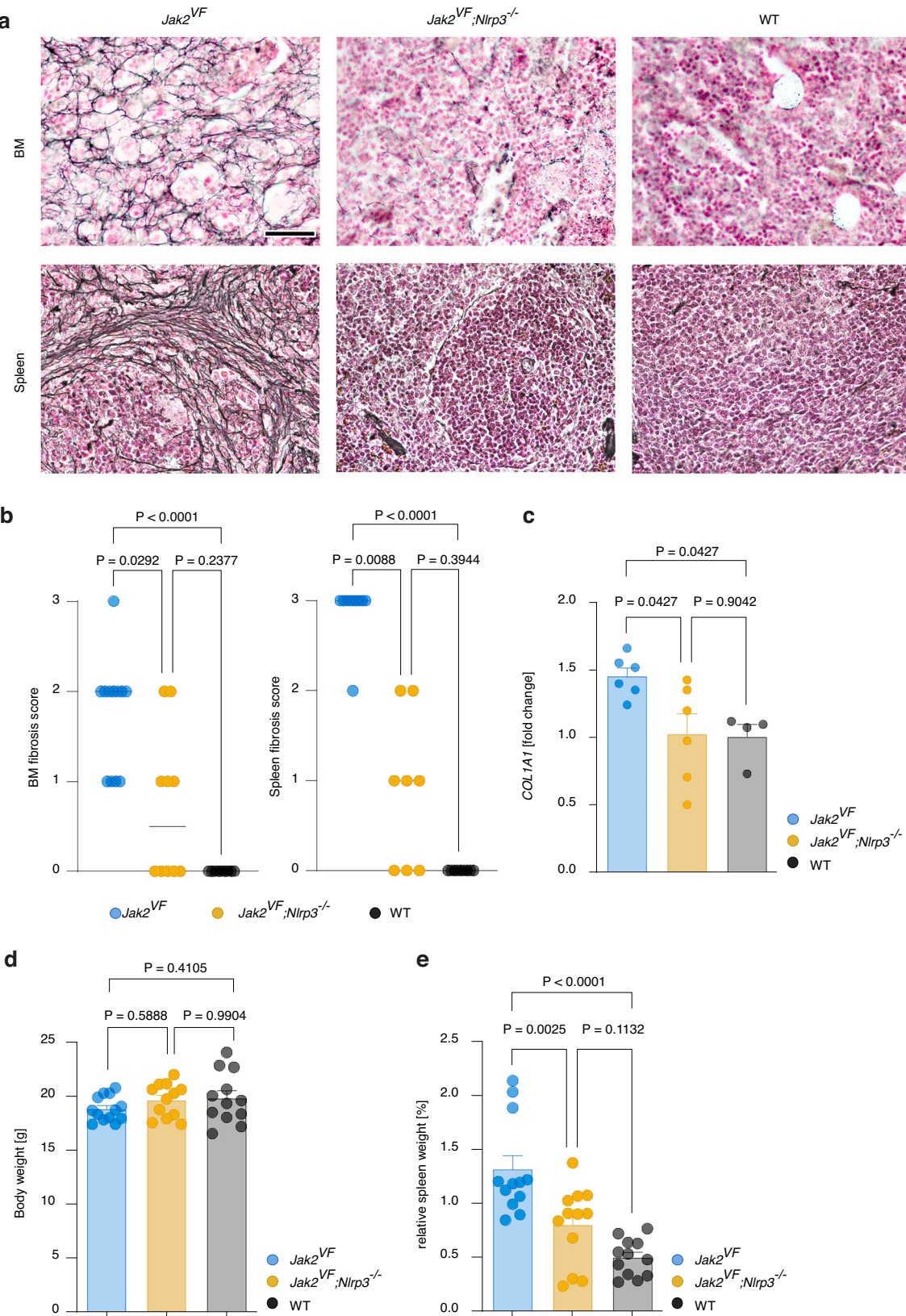

## Methods

### Primary human samples

MPN blood samples were obtained from patients enrolled in the German Study Group Myeloproliferative Neoplasms (GSG-MPN) bioregistry, approved by the Institutional Review Boards of the participating centers (DRKS-ID: DRKS00006035). Healthy control blood samples were drawn at the University Hospital of Bonn and collection was approved by the Ethics Committee of the Medical Faculty of the University of Bonn (154/13). Human studies were conducted according to the guidelines of the Declaration of Helsinki, and all subjects provided written informed consent before blood sampling.

**Fig. 7 | NLRP3 inflammasome activation propagates bone marrow fibrosis and splenomegaly. a** Representative images of silver-stained bone marrow (femur) and spleen sections of *Jak2^VF^*, *Jak2^VF^;Nlrp3^-/-^* and WT mice at 26 weeks of age for grading of fibrosis. Scale bar equals 50 μm. **b** Fibrosis of bone marrow (left) and spleen (right) was scored from 0 to 3. Shown are scores for bone marrow from *Jak2^VF^* (n = 12), *Jak2^VF^;Nlrp3^-/-^* (n = 10)⁻ and WT (n = 9) mice, and for spleen from *Jak2^VF^* (n = 10), *Jak2^VF^;Nlrp3^-/-^* (n = 8)⁻ and WT (n = 9) mice. Dots represent individual mice and horizontal lines the median. **c** Gene expression of *COL1A1* in bone marrow cells from *Jak2^VF^* (n = 6), *Jak2^VF^;Nlrp3^-/-^* (n = 6) and WT (n = 4) mice measured by qRT-PCR. Data represent normalized expression values relative to WT mice. **d** Body weight of *Jak2^VF^* (n = 13), *Jak2^VF^;Nlrp3^-/-^* (n = 12) and WT (n = 12) mice. **e** Relative spleen weight (percent of body weight) of *Jak2^VF^* (n = 12), *Jak2^VF^;Nlrp3^-/-^* (n = 12) and WT (n = 12) mice. Scatter bar plots in (**c–e**) show mean + SEM with dots representing individual mice. Statistically significant differences were determined by Kruskall-Wallis test with two-sided Dunn's multiple comparison test (**b**) and one-way ANOVA with two-sided Holm-Šidák multiple comparison test (**c–e**).

## Isolation and processing of human PBMCs

Whole blood was collected into EDTA tubes, and PBMCs were isolated by density gradient centrifugation with lymphocyte separation medium (PromoCell). PBMCs were either immediately used or cryopreserved at -80 °C in RPMI1640 complemented with 20% FCS and 10% DMSO. Monocytes were isolated from PBMCs using the Pan Monocyte Isolation Kit (Miltenyi).

## Mice

*Vav-Cre;Jak2^V617F/+^* C57BL/6 mice (*Jak2^VF^*) were generated and kindly provided by Jean-Luc Villeval[23]. *Jak2^VF^;Nlrp3^-/-^* mice were generated by interbreeding *Jak2^VF^* with *Nlrp3^-/-^* C57BL/6 mice (Millennium Pharmaceuticals). C57BL/6 J mice were purchased from Jackson Laboratories. Mice were maintained under special pathogen free (SPF) conditions, housed in individually ventilated cages (IVCs) in groups of no more than five animals. The housing rooms were standardized to a constant air temperature of approximately 22 °C, a humidity level of 50–60%, and up to 15 air changes per hour. The day-night cycle was centrally controlled and consisted of 12 h of light and 12 h of darkness. Experimental procedures were performed in accordance with the German Animal Welfare Act and approved by the State Agency for Nature, Environment and Consumer Protection, NRW (81-02.04.2017.A488, 81-02.04.2019.A427). Male and female mice were used in approximately equal numbers, except for the bone marrow transplantation recipients, which were all female.

## Bone marrow transplantation

10–12-week-old recipients were irradiated with $2 \times 4.5$ Gy in a 4 h interval and injected intravenously with $2.5 \times 10^6$ bone marrow cells harvested from 6–8-week-old donors. To generate *Jak2^VF^* BM, *Jak2^VF^;Nlrp3^-/-^* BM and WT BM mice, WT recipients were transplanted with 100% *Jak2^VF^*, *Jak2^VF^;Nlrp3^-/-^* or WT bone marrow ("neat" transplant). In all groups, care was taken to balance the ratio of female to male donors. Female mice were used as recipients.

## Preparation of single-cell suspensions

To prepare bone marrow cells, femurs were extracted and soft tissue removed with a scalpel and by gentle rolling over paper towels. Femurs were cut open on one side and placed into a 0.5 ml tube with a small hole at the tip, which was inserted into another 1.5 ml tube. Bone marrow cells were flushed from the bone by centrifugation. For splenocytes, spleens were removed, minced and pressed through a metal strainer. All cells were treated with Red Blood Cell Lysis buffer (Biolegend) and suspensions were filtered through 70 μm Nitex nylon mesh before further use.

## Sampling of mouse blood and treatment of mice

Blood was collected from the submandibular vein in live animals or from the abdominal vena cava after sacrifice. Blood counts were measured on an auto hematology analyzer (Mindray BC-5000 Vet) using EDTA-anticoagulated blood. In the pharmacological studies, mice were either fed a chow-based formulation of the NLRP3 inhibitor IFM-2384 (0.18 mg/g, IFM Therapeutics) or a control diet for 20 weeks.

## In vitro NLRP inflammasome activation

Cells were primed with LPS for 3 h or overnight at 37 °C with 100–200 ng/ml LPS in RPMI1640 containing penicillin, streptomycin

and 10% FCS. To activate NLRP3, nigericin (10 μM, Sigma Aldrich) was added to the cultures for 1 h. For detection of ASC specks, caspase-1 was inhibited with belnacasan (50 μM, Selleckchem) prior to stimulation with nigericin.

## Multiplex bead-based assays and ELISA

Human cytokines were measured with a custom ProcartaPlex Immunoassay (Thermo Fisher Scientific) and murine cytokines by LEGENDplex assay (Biolegend). The ProcartaPlex assay was read on a Luminex FlexMAP 3D System and LEGENDplex assays on a FACS Canto II (BD). Human and murine HMGB-1 were quantified by ELISA (IBL).

## Quantitative RT–PCR

RNA was extracted from cells with the Maxwell RSC simplyRNA Cells Kit (Promega) on a Maxwell RSC instrument and mRNA reverse transcribed with the RevertAid First Strand cDNA Synthesis Kit (Thermo Fisher Scientific). *GAPDH* was used as housekeeping gene. qPCR was performed using the SensiFast Sybr No-Rox Mix (Bioline) on a RealPlex 2 Thermocycler (Eppendorf).

Primer sequences (5'-3')
*GAPDH* forward GAGTCAACGGATTTGGTCGT
*GAPDH* reverse TTGATTTTGGAGGGATCTCG
*IL1B* forward TGGGCAGACTCAAATTCCAGCT
*IL1B* reverse CTGTACCTGTCCTGCGTGTTGA
*CASP1* forward ACAACCCAGCTATGCCCACA
*CASP1* reverse GTGCGGCTTGACTTGTCCAT

## Histology

Tissues were formalin-fixed, embedded in paraffin and sectioned (4–6 μm). Megakaryocytes were quantified in a blinded fashion on hematoxylin and eosin-stained sections. Fibrosis was scored in a blinded fashion on Gomori stained sections (0 = very little scattered linear reticulin with no intersections corresponding to normal BM, 1 = loose network of reticulin with few intersections, 2 = diffuse and dense increase in reticulin with extensive intersections, focal thick collagen fibers, 3 = coarse bundles of collagen fibers).

## Flow cytometry

Surface staining was performed on ice in PBS containing 0.5% BSA and 0.05% sodium azide. Intracellular staining was carried out using the FoxP3/Transcription Factor Staining Buffer Set (Thermo Fisher Scientific) or the Cytofix/Cytoperm Fixation/Permeabilisation Kit (BD). For exclusion of dead cells ethidium monoazide bromide (EMA) or ZOMBIE dyes (Biolegend) were used. Activated caspase-1/4/5 was detected with 5-FAM-YVAD-FMK kits (AAT Bioquest or ImmunoChemistry Technologies). Data were acquired on a Canto II (BD) or Northern Lights (Cytek) flow cytometer and analyzed with FlowJo 10.8.1 (BD).

## Antibody clones for flow cytometry

PE-conjugated anti-ASC (clone HASC-71, Cat.# 653904, mouse antibody, 1:100 dilution, Biolegend), PerCP/Cyanine5.5-conjugated anti-CD3 (clone 17A2, Cat.# 100218, rat antibody, 1:200 dilution, Biolegend), Pacific Blue-conjugated anti-CD11b (clone M1/70, Cat.# 101224, rat antibody, 1:200 dilution, Biolegend), Pacific Blue-conjugated anti-CD11c (clone N418, Cat.# 117322, Armenian hamster antibody, 1:200 dilution, Biolegend), PerCP/Cyanin 5.5-conjugated anti-CD19 (clone

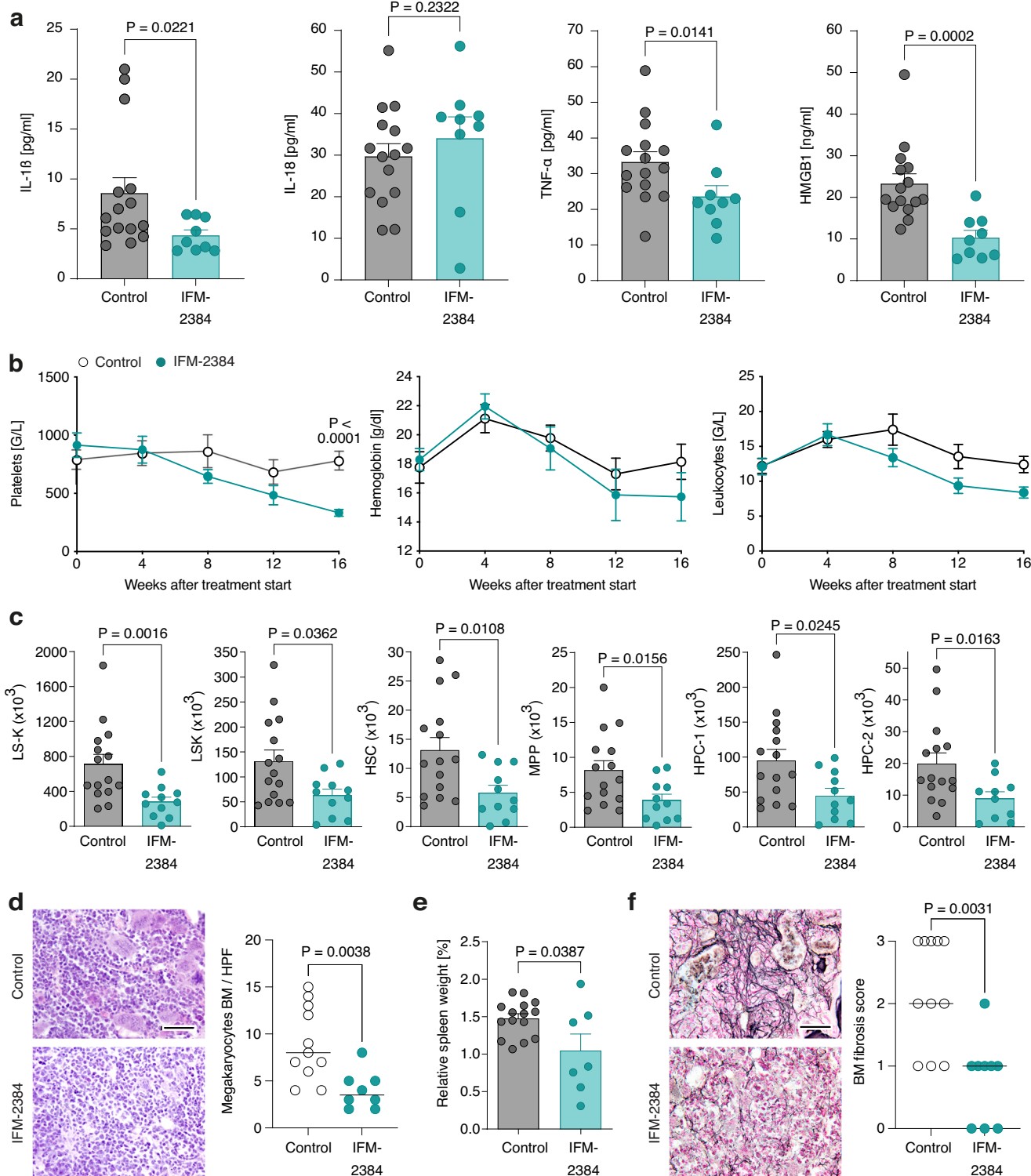

**Fig. 8 | Pharmacological blockade of NLRP3 alleviates thrombocytosis and splenomegaly.** *Jak2^VF* BM mice were fed for 20 weeks with chow containing the oral NLRP3 inhibitor IFM-2384 or control chow starting at 16 weeks after transplantation. **a** IL-1β, IL-18, TNFα and HMGB1 serum concentrations. IFM-2384 (n = 9), control chow (n = 15). **b** Blood counts. IFM-2384 (n = 11), control chow (n = 15). Blood was drawn by submandibular method. Plots show mean + SEM. **c** HSPC subset counts in the bone marrow. IFM-2384 (n = 11), control chow (n = 15). **d** Megakaryocyte counts (right) with representative images of H&E-stained bone marrow sections (left). Scale bar equals 50 μm. IFM-2384 (n = 8), control chow (n = 11). Dots represent individual mice and horizontal lines the median. **e** Relative

spleen weight (percent body weight). IFM-2384 (n = 7), control chow (n = 15). **f** Bone marrow fibrosis scores (right, scale 0 to 3) with representative images of silver-stained bone marrow sections (left). Scale bar equals 50 μm. IFM-2384 (n = 9), control chow (n = 11). Dots represent individual mice and horizontal lines the median. Scatter bar plots in (**a**, **c**, **e**) show mean + SEM with dots representing individual mice. Statistically significant differences were determined by two-tailed unpaired Mann-Whitney U test (**a**, **c**–**f**) and with two-tailed unpaired t test with two-stage linear step-up procedure of Benjamini, Krieger and Yekutieli (**b**). Source data are provided as a Source Data file.

6D5, Cat.# 115532, rat antibody, 1:200 dilution, Biolegend), FITC-conjugated anti-CD34 (clone RAM34, Cat.# 11-0341-82, rat antibody, 1:100 dilution, Invitrogen), Pacific Blue-conjugated anti-CD41 (clone MWReg30, Cat.# 133932, rat antibody, 1:200 dilution, Biolegend), PE-conjugated anti-CD42b (clone Xia.G5, Cat.# M040-2, rat-antibody, dilution 1:40, Emfret), Brilliant Violet 510-conjugated anti-CD48 (clone HM48-1, Cat.# 103443, Armenian hamster antibody, 1:200 dilution, Biolegend), APC-conjugated anti-CD117 (clone ACK2, Cat.# 135108, rat antibody, 1:100 dilution, Biolegend), PE-conjugated anti-CD150 (clone TC15-12F12.2, Cat.# 115904, rat antibody, 1:200 dilution, Biolegend), APC-conjugated anti-CD201 (EPCR) (clone RCR-16, Cat.# 351906, rat antibody, 1:200 dilution, Biolegend), Alexa Fluor 488-conjugated anti-GbVI (clone FAB6758G, Cat.# FAB6758G, rat antibody, 1:100 dilution, R&D), Pacific Blue-conjugated anti-F4/80 (clone BM8, Cat.# 123124, rat antibody, 1:400 dilution, Biolegend), PerCP-conjugated anti-Gr-1 (clone RB6-8C5, Cat.# 108426, rat antibody, 1:200 dilution, Biolegend), FITC-conjugated anti-ICAM (clone YN1/1.7.4., Cat.# 116105, rat antibody, 1:100 dilution, Biolegend), FITC-conjugated anti-Ki-67 (clone 16A8, Cat.# 652410, rat antibody, 1:400 dilution, Biolegend), APC-conjugated anti-Ly6C (clone HK1.4, Cat.# 128016, rat antibody, 1:800 dilution, Biolegend), PerCP-conjugated anti-Ly6G (clone 1A8, Cat.# 127654, rat antibody, 1:200 dilution, Biolegend), APC-conjugated anti-NLRP3 (clone REA668, Cat.# 130-111-210, human antibody 1:100 dilution, Milteny), FITC-conjugated anti-pro-IL-1β (clone NJTEN3, Cat.# 11-7114-82, rat antibody, 1:100 dilution, Invitrogen), FITC-conjugated rat IgG1κ isotype control antibody (clone RTK2071, Cat.# 400406, rat antibody, 1:100 dilution, Biolegend; used as control antibody in the pro-IL-1β staining), PE-Cy7-conjugated anti-Sca-1 (clone D7, Cat.# 25-5981-82, rat antibody, 1:300 dilution, Invitrogen), PerCP-conjugated, anti-Ter-119 (clone Ter119, Cat.# 116226, rat antibody, 1:200 dilution, Biolegend), Unconjugated anti-CD16/32 (Trustain, clone 93, Cat.# 101320, rat antibody, 1:400 dilution, Biolegend), PE-conjugated anti-VCAM (clone M/K-2, Cat.# RMCD10604, rat antibody, 1:40 dilution, Invitrogen).

### Generation of bone-marrow derived macrophages (BMDMs)
$7 \times 10^6$ bone marrow cells were cultured in tissue-culture-treated petri dishes in 8 ml of IMDM supplemented with glutamine, penicillin, streptomycin, 2-mercaptoethanol (all from Thermo Fisher Scientific), 10% FCS (Merck) and M-CSF (20 ng/ml, Biolegend). On day 4, 4 ml of fresh, prewarmed complete medium was added. On day 7, supernatants with non-adherent cells were discarded and adherent cells were carefully detached with a cell scraper and washed with PBS.

### Glycolysis/oxidative phosphorylation assay
Cell metabolism of bone marrow cells was assessed using the Glycolysis/OXPHOS assay Kit (Dojindo Laboratories) according the manufacturer's protocol.

### 3'-mRNA sequencing and gene expression analysis
For 3'-mRNA sequencing, RNA was extracted from sorted cells using the PicoPure RNA Isolation Kit (Thermo Fisher Scientific). Libraries were prepared with the Lexogen QuantSeq 3'-mRNA-Seq Library Prep Kit (FWD) and sequenced on an Illumina NovaSeq 6000 using single-end 100 bp reads. Raw data were pre-processed using Lexogen's web-based Kangooroo platform. Differential gene expression was analyzed in R (4.4.2) with DESeq2 (1.46.0), and gene set enrichment analysis (GSEA) was performed using clusterProfiler (4.14.6). Data visualization was carried out with ggplot2 (3.5.1) and enrichplot (1.26.6).

### 32D cell culture
32D cells (DSMZ, ACC 411) transduced with the pMSCV-MPL-HA-IRES-puromycin vector followed by a second transduction with the empty pMSCV-IRES-GFP vector or pMSCV-IRES-GFP containing the JAK2V617F,

CALRdel52 or CALRins5 oncogene were described previously[45]. 32D cells were cultured in RPMI 1640 medium with GlutaMAX (Gibco) supplemented with 10% FCS (Merck) and 1% penicillin/streptomycin (Gibco). For 32D MPL-HA cells without oncogene (empty second vector) the medium was additionally supplemented with 5 ng/ml murine IL-3 (Thermo Fisher Scientific). All cells were maintained at 37 °C in a humidified incubator with 5% $CO_2$.

### Caspase 1 bioluminescent assay
Caspase 1 activation in 32D cells was measured with the Caspase-Glo 1 Inflammasome Assay (Promega) according to the manufacturer's protocol. Luminescence was measured on a GloMax Multi+ Detection System (Promega).

### Western blot
Whole cells lysates from BMDMs were prepared in RIPA buffer (Sigma-Aldrich) containing phosphatase inhibitor cocktails 2&3 (Sigma-Aldrich). Supernatant proteins were precipitated with chloroform/methanol. Protein concentrations were determined by bicinchoninic acid assay (Pierce). 40 μg of whole-cell lysates or precipitates of 500 μl supernatant were separated on a poly-acrylamide gel and transferred on poly-vinyl dinitrofluoride (PVDF) membranes. PVDF membranes were then immunoprobed with antibodies against murine IL-1ß (DY401, R&D) and ß-Actin (#4967, Cell Signaling) followed by HRP-labeled secondary antibodies. Proteins were detected with an ECL system (GE Healthcare) on a ChemiDoc XRS+ (Bio-Rad).

### Quantification and statistical analysis
Statistical analysis was performed with GraphPad 9 (Prism). The statistical tests used are indicated in the figure legends. Independent experiments were pooled and analyzed together whenever possible. Clustering analysis and visualization were performed using the ComplexHeatmap package (2.14.0) in R (4.4.2).

### Reporting summary
Further information on research design is available in the Nature Portfolio Reporting Summary linked to this article.

## Data availability
The raw data for charts and graphs, and unedited blot images are available in the Source Data. 3'-mRNA sequencing data have been deposited in the Zenodo repository and can be accessed via https://doi.org/10.5281/zenodo.16893730[46]. Source data are provided with this paper.

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

## Acknowledgements

We thank HET (Haus für Experimentelle Therapie) for outstanding animal husbandry, Radek Skoda and Alexander Gerbaulet for helpful discussion

and critical reading of the manuscript, and the GSG-MPN centers for patient samples. This study was funded by the Deutsche Krebshilfe (DKH, German Cancer Aid) Mildred Scheel Early Career Center grant 70113307 (R.M.K.) and the Deutsche Forschungsgemeinschaft (DFG, German Research Foundation) grant TRR237-369799452 (L.L.T.). It was further supported by the DFG grant EXC2151–390873048 (L.L.T.) under Germany's Excellence Strategy, the Jose-Carreras-Leukämie Stiftung grant 16 R/2018 (D.W.), and in part by DFG grants 417911533 (THB) and 327211770 (SK) within the clinical research unit CRU344.

## Author contributions

R.M.K. and L.L.T. designed experiments, analyzed the data and wrote the manuscript. R.M.K., C.K., K.C., M.L.S., E.T. and C.K.B. performed experiments. C.C.K. performed the Luminex assay. T.H.B., S.K. and M.G. provided patient samples. I.G. helped with histology. P.B., E.L. and T.F. provided resources. D.W. and L.L.T. conceived and supervised the study. All authors had the opportunity to discuss the results and comment on the manuscript.

## Funding

## Competing interests

R.M.K. received honoraria from Stemline. C.K. is an employee of AbbVie Deutschland GmbH & Co. KG. T.H.B. served as a consultant or invited speaker for Astra-Zeneca, Gilead, Janssen, Merck, Novartis and Pfizer and received research funding from Novartis and Pfizer. SK reports research grant/funding from Geron, Janssen, AOP Pharma, and Novartis; consulting fees from Pfizer, Incyte, Ariad, Novartis, AOP Pharma, Bristol Myers Squibb, Celgene, Geron, Janssen, CTI BioPharma, Roche, Bayer, GSK, Sierra Oncology, and PharmaEssentia; payment or honoraria from Novartis, BMS/Celgene, Pfizer; travel/accommodation support from Alexion, Novartis, Bristol Myers Squibb, Incyte, AOP Pharma, CTI Bio-Pharma, Pfizer, Celgene, Janssen, Geron, Roche, AbbVie, GSK, Sierra Oncology, and Karthos; a patent issued for a BET inhibitor at RWTH Aachen University; advisory board activity for from Pfizer, Incyte, Ariad, Novartis, AOP Pharma, BMS, Celgene, Geron, Janssen, CTI BioPharma, Roche, Bayer, GSK, Sierra Oncology, and PharmaEssentia. MG reports speaker bureau and consultancy for AOP Orphan, Celgene, CTI, Novartis, and Shire. I.G. received funding from aTyr. E.L. is co-founder and consultant to IFM Therapeutics, Odyssey Therapeutics and a Stealth Biotech company. D.W. served as a consultant or invited speaker for Novartis, Roche, BMS, Gilead, Janssen, MSD, AOP Orphan and Pfizer and received research funding from Novartis, BMS, AOP Orphan and Pfizer. L.L.T. reports honoraria from AOP Orphan, Boehringer Ingelheim, BMS and Novartis; consultancy for Astellas, Blueprint, BMS, GSK, Jazz Pharmaceuticals, Pfizer, and Sobi. The remaining authors declare no competing financial interests.

## Additional information

[1]Department of Medicine III, University Hospital Bonn, Bonn, Germany. [2]Mildred Scheel School of Oncology, University Hospital Bonn, Medical Faculty, Bonn, Germany. [3]Center for Integrated Oncology Aachen Bonn Cologne Düsseldorf (CIO ABCD), Bonn, Germany. [4]Department of Hematology, Oncology, Hemostaseology, and Stem Cell Transplantation, Faculty of Medicine, RWTH Aachen University, Aachen, Germany. [5]Center for Integrated Oncology Aachen Bonn Cologne Düsseldorf (CIO ABCD), Aachen, Germany. [6]University Clinic for Hematology, Oncology, Haemostaseology and Palliative Care, Johannes Wesling Medical Center Minden, University of Bochum, Bochum, Germany. [7]Institute of Pathology, University Hospital Bonn, Bonn, Germany. [8]Institute of Molecular and Clinical Immunology, Otto-von-Guericke University of Magdeburg, Magdeburg, Germany. [9]Institute of Innate Immunity, University Hospitals Bonn, Bonn, Germany. [10]Deutsches Rheuma Forschungszentrum Berlin (DRFZ), Berlin, Germany. [11]Internal Medicine V, Department of Hematology and Oncology, Comprehensive Cancer Center Innsbruck (CCCI), Tyrolean Cancer Research Institute (TKFI), Austrian Comprehensive Cancer Network (ACCN), Medical University of Innsbruck, Innsbruck, Austria. [12]These authors contributed equally: Dominik Wolf, Lino L. Teichmann. ✉e-mail: dominik.wolf@i-med.ac.at; lino.teichmann@uni-bonn.de

