## [Transparent Peer Review file · Nature Communications]

NLRP3-induced systemic inflammation controls the development of JAK2V617F mutant myeloproliferative neoplasms

Corresponding Author: Dr Lino Teichmann

Version 0:

Reviewer comments:

Reviewer #1

(Remarks to the Author)

In this study, Koerber and colleagues present a well-conducted study of the role of the Nlrp3 inflammasome in myeloproliferative neoplasms. The authors describe the contribution of NLRP3, which is activated downstream of a plethora of inflammatory signals, to the progression of JAK2V617F-driven MPN in a murine model. They subsequently use of IFM-2384, a novel NLRP3 inhibitor to normalise the blood phenotype, splenomegaly and bone marrow fibrosis and describe this as a potential novel therapeutic approach to treat MPNs. Evidence for NLRP3 involvement in MPN patients was first presented in 2020 by Zhou and colleagues, and has been hinted at even prior, so the novelty of the importance of NLRP3 in MPN patients is a little lost. However, the use of the IFM2384 inhibitor is a novel and interesting therapeutic option, which warrants further research. Overall, the subject is interesting and pertinent, and the experiments are carefully designed and rigorous. However, the authors are encouraged to consider the following points to improve their manuscript and strengthen their conclusions.

- 1) It is interesting to this reviewer that there is so much heterogeneity in the patient data shown in figure 1a, which does not appear to either depend on driver mutation or therapeutic regimen received – considering less than half of patients examined cluster together in the IL1b/Il18 high cluster, could the authors suggest why that may be the case and whether there are additional parameters that could explain these differences?
- 2) It is unclear from the methods/results section how the JAK2VF and JAK2VFxNlrp3KO mice were transplanted into recipients – is it a “neat” transplant, or are these donors cells mixed in a particular ratio? Do they consistently develop a phenotype (ie. Thrombocytopenia, fibrosis)? This has been a longstanding issue in the field – Along this line, are the authors able to recapitulate similar findings in other JAK2 knockin/retroviral transduced/TPO overexpression/CALRdel52/MPLW515L models of MPN? This would certainly strengthen the authors’ claims that NLRP3 is a key player in MPN development.
- 3) The data highlights nicely that NLRP3 plays a role in MPN, as has been hinted at over the last decade. It is now better described and clearly an important downstream pathway in the disease pathogenesis. However, the authors mention throughout the manuscript that NLRP3 drives disease progression – have the authors checked whether NLRP3 overexpression/overactivation in WT mice leads to some features of MPN (fibrosis, splenomegaly/cytopenias)?
- 4) Could the authors elaborate as to why pyroptosis is increased in JAK2VF HSPCs compared to JAK2xNlrp3ko mice but also observed increased HSPC numbers?
- 5) It is intriguing that Nlrp3 deficiency suppresses platelet production and decreases MK numbers, yet do not see a reduction in TPO levels. Is there an explanation for this? Does NLRP3 deficiency also attenuate MK dysplasia, which is also a major hallmark, or simply reduce numbers?
- 6) A major concern remains that NLRP3 is involved in HSC mobilization, and its deficiency or inhibition seems to have a

strong effect on the HSC compartment. Have the authors checked the functional effect of NLRP3 deficiency and/or inhibition on the HSC compartment? Specifically, is the reduction observed specifically in mutated cells or does it also impact WT HSCs? Do intact WT HSCs remain functional after treatment?

7) The use of the NLRP3 inhibitor IFM-2384 is very promising, particularly with regards to the prevention of bone marrow fibrosis, yet it is unclear how this might be the case? Does the inhibitor specifically act on mutated hematopoietic cells? Does it alter ligand-receptor crosstalk with the stromal compartment? Does knocking out NLRP3 in the stromal compartment also lead to an amelioration of fibrosis/splenomegaly (albeit perhaps not the blood phenotype)?

8) An important issue with current MPN treatments, specifically with Ruxolitinib, is the development of severe cytopenias, which ultimately results in patients coming off of the drug. Have the authors tested the synergistic effect of Rux and IFM2384 on human cells/HSPCs (ie. CFU assays)?

9) Which cell populations specifically drive this NLRP3 expression/disease progression? Is this primarily neutrophils?

Minor comments

The figures are generally well designed, but please include individual data points to provide more transparency.

Line 230 – typo "capase-1"

Reviewer #2

(Remarks to the Author)

Earlier studies (Refs 9,10) have shown that inhibition of IL-1b or IL-1 signaling reduced platelet counts and features of myelofibrosis in Jak2VF mice. The present work extends these findings by showing that these changes are being driven in major part by NLRP3 inflammasome activation in chow fed mice. They also provide new mechanistic insights into the role of the NLRP3 inflammasome in driving the short route from HSCs to MkPs, although there is limited insight into whether this is a direct effect of IL-1b in the LSK population and if so the changes that are involved. Importantly, they show that an NLRP3 inhibitor reduces platelet counts and features of myelofibrosis. Overall, the findings are novel and interesting, the studies are carefully performed and the paper is well written.

Specific Comments:

1. Ref 34 already reported increased NLRP3 inflammasome activation in murine and human JAK2VF cells, and showed increased mitochondrial ROS generation as a potential underlying mechanism. This does not detract from the present report but needs to be acknowledged.
2. The evidence for pyroptosis of HSCs is based on indirect measurements. Can the authors use an antibody to the N-terminal fragment of GSD to confirm these findings?
3. What is the mechanism by which NLRP3 influences the HSC-Mkp axis?
4. Experiments done with the NLRP3 inhibitor (IFM) lack a WT control group. In view of the therapeutic potential it would be important to at least show no effect on platelets in WT controls.
5. Line 151. By "real world conditions" the authors mean uncontrolled observational conditions. Previous reports have shown that Rux decreases IL-18 and increases LDL levels under conditions where the patient is compared before and after treatment.

Reviewer #3

(Remarks to the Author)

Koerber et al., in this manuscript, explored the roles of NLRP3 inflammasome in JAK2VF MPN disease development and potential therapy. Through cytokine profiling in clinical samples, they found that IL-1 β and IL-18, which were closely associated with the activation of NLRP3 inflammasome, were significantly higher in MPN patients than in healthy individuals. Using JAK2V617F knock-in mice, the authors found that loss of NLRP3 attenuated MPN phenotypes, including platelet and white blood cell counts in peripheral blood and hematopoietic stem and progenitor cell (HSPC) subpopulation numbers in the bone marrow via decreased IL-1 β and IL-18 secretion. Furthermore, the authors found that NLRP3 regulates thrombopoiesis and promotes fibrosis and splenomegaly in JAK2V617F knock-in mice. Finally, blocking NLRP3 with the novel oral inhibitor was found to have potential therapeutic effectiveness in treating MPN. The authors claimed that NLRP3 is critical for MPN development, and its inhibition represents a new therapeutic intervention for MPN patients. My major comments are listed below:

1. NLRP3/Caspase 1/IL-1 β pathway is well-known in many inflammation-mediated diseases, including myeloproliferative neoplasms (Shivam Rai et al., Nature Communications, 2022, Mohammed Ferdous-Ur Rahman et al., Nature Communications, 2022). NLRP3's roles shown in this paper are very similar to those of IL-1 β -IL1R1 in those two published papers. Comparing NLRP3 and IL-1 β side by side in many assays to show NLRP3's unique roles may strengthen the novelty of this study.
2. Can the authors perform a competitive bone marrow transplant experiment to determine whether NLRP3 inhibitor affects JAK2V617F allele burden? It would be interesting to explore the combination of Ruxolitinib and NLRP3 inhibitor. Figure 1g showed that Ruxolitinib did not affect IL-1 β and IL-18 secretion. Combination with NLRP3 inhibitor may avoid Ruxolitinib resistance.
3. The authors showed that the loss of NLRP3 impairs platelet production via the short route of HSCs to MkPs. However,

how NLRP3 regulates thrombopoiesis is unclear. RNA-seq or proteomic assay may help provide some clues.

4, The authors showed in Figure 4a that CMPs, GMPs, MEPs, and MkPs were diminished in Jak2VF Nlrp3^{-/-} BM relative to Jak2VF BM mice. However, these figures were not shown. It is also unclear whether other HSPC subpopulations were affected.

5, Figure 6c is not convincing. The Jak2VF Nlrp3^{-/-} group had 9 mice in the body weight panel, and there were 8 mice in the Jak2VF Nlrp3^{-/-} group in the relative spleen weight panel. What would the statistics be like when all the mice are included? A similar case was found in Figure 6b. These figures are not convincing and should be revised.

Reviewer #4

(Remarks to the Author)

The work by Koerber and colleagues is focused on the role of inflammasome activation in MPN patients and JAK2VF murine cells and explores the possible benefit of inhibiting this pathway by using genetic and pharmacologic approaches. They demonstrate elevated levels of inflammasome-processed cytokines IL-1 β and IL-18 in patient and murine serum, which are reduced in the absence of NLRP3. Inflammasome activation is suggested by increased ASC speck formation in patient cells and the release of mature IL-1 β from JAK2VF macrophages following NLRP3 stimulation with LPS+Nigericin. NLRP3 ablation or pharmacologic inhibition reduced platelet, and to a lesser extent, leukocyte counts with no changes in hemoglobin levels, reduced HSCP and MK numbers, BM and spleen fibrosis and splenomegaly, pointing to NLRP3 as a therapeutic target in MPN, which could be of potential clinical interest. The work is well performed and novel. However, insight into the signals leading to NLRP3 activation as well as the mechanism underlying the effect of NLRP3 on the myeloproliferative phenotype are limited, which diminishes the impact of the findings.

Comments:

1-Figure 7D seems to be missing. Please include it.

2- Evidence for enhanced inflammasome activation in patient cells (mild increase in ASC specks by flow cytometry) should be reinforced by additional experiments, such as for example measurement of caspase 1 activation or further confirmation of the presence of ASC specks by another method. Besides, it would be relevant to further demonstrate the specific participation of NLRP3, for example by showing colocalization of NLRP3 in ASC specks by immunofluorescence and/or blockade of ASC speck formation by using a specific NLRP3 inhibitor. Besides, the authors state that ASC specks were increased in peripheral blood "myeloid" cells: are these neutrophils? monocytes? myeloid precursors? how were these cells identified?

3-The mechanism underlying inflammasome hyperactivation in MPN is not entirely clear. The authors suggest it might be due to JAK2-dependent inflammasome priming. Elevated CASP1 and IL-1 β in patient cells is shown in Figure 2A. Considering the work is overall focused on NLRP3, the authors should also show NLRP3 levels in these samples and analyze whether there are differences between JAK2-positive and -negative patients. NLRP3 levels could also be assessed in the murine JAK2VF model. To this reviewer's knowledge, NLRP3 has not been described as a downstream JAK2 target.

4-Following on this comment, could high TNF α levels contribute to NLRP3 inflammasome activation in the JAK2VF model? It could be of interest to test whether NLRP3 activation is ameliorated by the use of TNF blocking antibodies or inhibitors.

5- JAKVF NLRP3^{-/-} mice show lower platelet/leukocyte counts vs. JAKVF and WT mice. Blood counts of NLRP3^{-/-} mice should be shown for comparison to exclude this is due to the effect of NLRP3 ablation on normal hematopoiesis.

6- The authors state that NLRP3 ablation reduces platelet counts at least in part by inhibiting the direct thrombopoiesis pathway, as shown by lower levels of CD48-low MKs. They suggest this feature could be due to the reduction in IL-1 β . Does IL-1 β regulate the direct thrombopoiesis pathway? Further experiments should be performed to support this statement, for example evaluating this pathway in JAK2VF with IL-1 β ablation or pharmacologic IL-1 β inhibition; alternatively the effect of recombinant IL-1 β in wild-type mice would also be informative.

7- The reference list does not comprehensively cover the available literature: a) Elevation of inflammasome-processed cytokines IL-1 β (PMID: 36100613, PMID: 36100596) and IL-18 (PMID: 38035089) has been previously described in in plasma/serum from MPN patients. The authors should refer to these previous findings. b) Enhanced NLRP3 inflammasome activation was previously shown in LPS and Nigericin-stimulated monocytes from patients with Myelofibrosis (PMID: 38035089). Please acknowledge these findings in the manuscript.

Minor comments

-Please explain why TNF α is reduced by NLRP3 deletion/inhibition in the JAKVF model (Figure 2D and Figure 7A)?

- Please detail how were MkP identified (Figure 5C).

-It is surprising that differences in TPO levels in the different animal groups were not significant, as data look quite different (Figure 5D).

- In Figure 1E, 2A and 2B, please detail which MPN patients were included in each assay (number of ET, PV and PMF patients).

-Figure 1 A data are shown in two separate sets. Please explain how were these patients divided in the figure legend.

- Figure 2 A y B. Please provide captions for the colour code in the graphs (do gray bars represent HC and red bars, patients?)

Version 1:

Reviewer comments:

Reviewer #1

(Remarks to the Author)

The revision experiments and added data have significantly improved the manuscript, and I am satisfied with the manuscript.

Reviewer #2

(Remarks to the Author)

The revised paper has been improved by some additional data and a more balanced discussion. The study points to the potential use of NLRP3 inhibitors in JAK2VF MPN papers and provides some insights into mechanisms underlying decreases in platelet counts.

Reviewer #3

(Remarks to the Author)

My first major concern pertains to the novelty of the manuscript. The role of NLRP3 in inflammatory diseases is well established, and its involvement in MPNs has been previously reported. Therefore, without the discovery of significant new insights regarding the detailed roles of NLRP3 in MPNs, the manuscript does not present a sufficiently novel contribution.

In addition, the authors did not adequately address my second major concern, whether inhibition of NLRP3 can reduce the JAK2V617F allele burden. It is unclear why this would require 1–2 years to complete, as competitive transplantation experiments are standard and routinely performed in the field.

Reviewer #4

(Remarks to the Author)

The authors have adequately answered the comments raised by the reviewer

Point-by-point reply

We would like to thank the reviewers for their thoughtful and thorough evaluation of our manuscript. We are grateful for their positive feedback, particularly regarding the new insights our study offers and the overall quality of the work, despite some limitations.

We acknowledge the concerns raised by the reviewers. Many of the issues could be addressed by additional clarification and modification. In addition to addressing these points, we recognized the need for additional data to strengthen the main findings of the paper. As such, we have conducted further experiments, and the results have been incorporated into the revised manuscript. Below, we provide detailed responses to each of the reviewers' comments.

In our Point-by-Point reply, we highlighted in yellow instances where we made changes to the manuscript text or added new data. Corresponding changes in the manuscript are also highlighted in yellow. Please note that the figure legends were substantially revised due to the requested replacement of bar plots with scatter bar plots, which required adjustments to the descriptions (e.g., clarifying that each dot represents an individual patient), as well as the addition of figures and reorganization of some figure panels. These type of updates to the figure legends are not highlighted.

Reviewer #1:

In this study, Koerber and colleagues present a well-conducted study of the role of the Nlrp3 inflammasome in myeloproliferative neoplasms. The authors describe the contribution of NLRP3, which is activated downstream of a plethora of inflammatory signals, to the progression of JAK2V617F-driven MPN in a murine model. They subsequently use of IFM-2384, a novel NLRP3 inhibitor to normalise the blood phenotype, splenomegaly and bone marrow fibrosis and describe this as a potential novel therapeutic approach to treat MPNs. Evidence for NLRP3 involvement in MPN patients was first presented in 2020 by Zhou and colleagues, and has been hinted at even prior, so the novelty of the importance of NLRP3 in MPN patients is a little lost. However, the use of the IFM2384 inhibitor is a novel and interesting therapeutic option, which warrants further research. Overall, the subject is interesting and pertinent, and the experiments are carefully designed and rigorous. However, the authors are encouraged to consider the following points to improve their manuscript and strengthen their conclusions.

We thank the reviewer for the positive feedback and for highlighting the rigor and careful design of our experiments. We appreciate the recognition of the novelty and potential of the IFM-2384 inhibitor as a therapeutic option.

Regarding the important work by Zhou et al.¹, this study primarily focused on the expression of NLRP3 and did not address its functional role in disease development. In contrast, our work provides a deeper understanding by demonstrating the mechanistic involvement of NLRP3 in the pathogenesis of MPN. Thus, while Zhou's study identified

NLRP3 as a relevant marker, our study expands on this by defining its role in disease development and offering a potential therapeutic strategy. **To acknowledge Zhou's work, we now cite it in the Introduction.**

1) It is interesting to this reviewer that there is so much heterogeneity in the patient data shown in figure 1a, which does not appear to either depend on driver mutation or therapeutic regimen received – considering less than half of patients examined cluster together in the IL1b/IL18 high cluster, could the authors suggest why that may be the case and whether there are additional parameters that could explain these differences?

There is indeed substantial variability in the patient data presented in Figure 1a. While we did not specifically address certain factors that could influence this heterogeneity, we recognize that a number of important variables could contribute. First, our data reflect a single time point, which may not fully reflect the average serum concentrations that might be present in a patient over a longer period of time. Moreover, variations in drug dosing, including suboptimal doses in some patients, could further contribute to the observed variability. Another factor is that some patients were in earlier treatment lines and others in later lines. Further complicating the analysis is the potential impact of age and sex. We also did not explore specific risk groups, such as high-risk versus low-risk PMF, which could have added an additional layer of stratification and potentially clarified some of the observed heterogeneity.

These factors were not specifically examined in Figure 1a because it was not the primary focus and because clarifying their role would require a greater sample size.

Despite this variability, it is important to note that we were able to distinguish different cytokine patterns between groups. This ability to discern meaningful differences, even in the presence of substantial variability, arguably underscores the robustness of the finding.

2) It is unclear from the methods/results section how the JAK2VF and JAK2VFxNlrp3KO mice were transplanted into recipients – is it a “neat” transplant, or are these donors cells mixed in a particular ratio? Do they consistently develop a phenotype (ie. Thrombocythemia, fibrosis)? This has been a longstanding issue in the field – Along this line, are the authors able to recapitulate similar findings in other JAK2 knockin/retroviral transduced/TPO overexpression/CALRdel52/MPLW515L models of MPN? This would certainly strengthen the authors’ claims that NLRP3 is a key player in MPN development.

It is a “neat” transplant i.e. lethally irradiated WT mice were transplanted with 100% Jak2VF, 100% Jak2VF;Nlrp3^{-/-} or 100% WT bone marrow. **We explicitly mention this now in the Methods section.** The consistency of the phenotype of the transplanted mice

(and the non-manipulated, non-transplanted mice) can be assessed by the scatter characteristics and the error bars in Figure 3-8.

Regarding the use of additional mouse models, we would kindly ask the reviewer to consider the following points:

Our study specifically investigates JAK2V617F mutant MPNs, as stated in the title. JAK2V617F is the most common driver mutation in MPNs, present in 95% of polycythemia vera (PV) patients and 50% of essential thrombocythemia (ET) and primary myelofibrosis (PMF) patients. Given this focus, we believe it is more relevant to strengthen our findings within this context rather than extend them to other driver mutations (such as CALRdel52 or MPLW515L) or to conditions like TPO overexpression, which leads to a different type of MPN.

To enhance the robustness of our conclusions, we have employed several key strategies:

- Genetic and pharmacological targeting of NLRP3: We used two independent approaches (genetic and pharmacological) to inhibit NLRP3. This allowed us to inhibit NLRP3 both before and after disease onset, allowing for a comprehensive evaluation of its role.
- Control experiments using unmanipulated JAK2V617F mice: In addition to transplantation of JAK2V617F bone marrow into wild-type recipients, we included unmanipulated JAK2V617F mice to account for potential confounding effects from the transplantation procedure.
- Most importantly, our study incorporates data from clinical samples collected from MPN patients, which strongly support the involvement of the NLRP3 inflammasome in human MPN.

Finally, we must adhere to EU regulations on animal research, particularly the 3Rs principle (Replacement, Reduction, and Refinement). Repeating our experiments in additional mouse strains would likely not be approved by the State Agency for Nature and Environmental Protection of the state of NRW.

However, it is worth pointing out that in the murine myeloid cell line 32D we found that caspase 1 activation is not only induced by JAK2V617F, but also by other MPN driver mutations, including CALRdel52 and CALRins5.

The data are shown Figure 3c.

3) The data highlights nicely that NLRP3 plays a role in MPN, as has been hinted at over the last decade. It is now better described and clearly an important downstream pathway in the disease pathogenesis. However, the authors mention throughout the manuscript that NLRP3 drives disease progression – have the authors checked whether NLRP3

overexpression/overactivation in WT mice leads to some features of MPN (fibrosis, splenomegaly/cytopenias)?

Gain-of-function mutations in NLRP3, as shown in previous studies², lead to early mortality, which complicates the study of fibrotic progression in such models. However, in those models, NLRP3 overactivation results in splenomegaly and leukocytosis. Interestingly, some of these features appear to be partly independent of IL-1 β and IL-18³.

The downstream effector cytokine IL-1 β has been shown to promote the maturation of megakaryocytes into cells with higher ploidy and to increase platelet adhesion and the formation of heterotypic aggregates in both high-fat diet and bacterial infection in the WT mouse setting⁴.

4) Could the authors elaborate as to why pyroptosis is increased in JAK2VF HSPCs compared to JAK2xNlrp3ko mice but also observed increased HSPC numbers?

To clarify the observed increase in HSPCs in the presence of increased pyroptosis in Jak2VF BM mice compared to Jak2VF;Nlrp3^{-/-} BM mice, we assessed cell proliferation by staining for Ki67. Our data show that NLRP3 promotes HSPC proliferation. We interpret these findings to mean that while NLRP3 influences pyroptosis, its effect on cell proliferation is more pronounced, leading to an overall increase in HSPC numbers despite the elevated levels of pyroptosis.

The Ki67 data is now included in Figure 5e.

5) It is intriguing that Nlrp3 deficiency suppresses platelet production and decreases MK numbers, yet do not see a reduction in TPO levels. Is there an explanation for this? Does NLRP3 deficiency also attenuate MK dysplasia, which is also a major hallmark, or simply reduce numbers?

In Figure 6e (former Figure 5d), we show that TPO levels are reduced in Jak2VF mice compared to wild-type mice, which confirms the well-established reciprocal relationship between TPO serum concentrations and platelet count⁵. A similar, but better known, negative feedback loop exists between erythropoietin and hemoglobin levels (a suppressed erythropoietin concentration is a diagnostic criterion in establishing the diagnosis of polycythemia vera). Based on this, one might expect that the lower platelet counts observed in NLRP3-deficient mice would lead to an increase in TPO levels. Indeed, Figure 6e shows a trend towards higher TPO concentrations in Jak2VF;Nlrp3^{-/-} mice compared to Jak2VF mice, although this effect did not reach statistical significance.

What this implies is that NLRP3 is able to promote thrombopoiesis by a mechanism that does not require TPO, which aligns with studies demonstrating a direct effect of IL-1 β on megakaryopoiesis (as discussed in response to Comment 3).

Regarding dysmegakaryopoiesis, we analyzed cell size and nuclear morphology for the revised manuscript. We found that deletion of Nlrp3 in Jak2VF mice reduces megakaryocyte size and dysplastic features of the nuclei. **The data are now included in Figure 6d.**

6) A major concern remains that NLRP3 is involved in HSC mobilization, and its deficiency or inhibition seems to have a strong effect on the HSC compartment. Have the authors checked the functional effect of NLRP3 deficiency and/or inhibition on the HSC compartment? Specifically, is the reduction observed specifically in mutated cells or does it also impact WT HSCs? Do intact WT HSCs remain functional after treatment?

In our mouse models, all hematopoietic cells were JAK2V617 mutant. Consequently, we were unable to assess the specific effects of NLRP3 deficiency or inhibition on wild-type (WT) hematopoietic stem and progenitor cells (HSPCs). However, these effects have been explored by other research groups, particularly in a series of studies by the Ratajczak lab.

The Ratajczak lab has reported that NLRP3 promotes HSPC mobilization in response to stimuli such as G-CSF or nigericin, using LSKs and CFU-GM colonies in the blood as primary readouts⁶. Notably, HSCs were not directly quantified in these studies. The relevance of this finding outside the context of stem cell collection for transplantation purposes remains unclear.

Additionally, the same group observed slower engraftment of NLRP3-deficient bone marrow into irradiated hosts compared to WT bone marrow⁷. However, this delay was transient, as no differences were observed in engraftment after 28 days, implying that NLRP3 deficiency does not result in long-term impairment of HSC function.

Somewhat in contrast, Luo et al. demonstrated that NLRP3 inflammasome activation contributes to the functional decline of HSCs during aging⁸. Their work suggests that inhibition of NLRP3 can actually improve HSC regenerative capacity, rather than impairing it.

Taken together, these findings suggest that NLRP3 deficiency or inhibition does not profoundly compromise HSC function and, in some contexts, may even enhance it.

7) The use of the NLRP3 inhibitor IFM-2384 is very promising, particularly with regards to the prevention of bone marrow fibrosis, yet it is unclear how this might be the case? Does the

inhibitor specifically act on mutated hematopoietic cells? Does it alter ligand-receptor crosstalk with the stromal compartment? Does knocking out NLRP3 in the stromal compartment also lead to an amelioration of fibrosis/splenomegaly (albeit perhaps not the blood phenotype)?

To investigate whether IFM-2384 acts directly on mutated hematopoietic cells, we performed additional transplant experiments. Jak2VF-mutated bone marrow cells were transplanted into both *Nlrp3*^{-/-} and wild-type (WT) mice. After 12 weeks, femurs were harvested and analyzed for the development of bone marrow fibrosis. The results indicate that the NLRP3 status in radioresistant non-hematopoietic cells (including stromal cells) does not influence fibrosis development. This suggests that the reduction in bone marrow fibrosis observed in Jak2VF BM mice treated with IFM-2384 is primarily due to the inhibitor's action on mutated hematopoietic cells, rather than on the stromal compartment. **The data is presented in Figure S1c.**

Regarding whether IFM-2384 ameliorates bone marrow fibrosis by altering ligand-receptor interactions with the stromal compartment, our data suggest that the observed effects are likely related to changes in megakaryocyte numbers. In our experiments, NLRP3 inhibition in Jak2V617F mutant mice substantially reduced the number of megakaryocytes. This is significant because megakaryocytes play a key role in promoting bone marrow fibrosis. Specifically, megakaryocytes in myelofibrosis are known to release increased amounts of pro-fibrotic cytokines, which contribute to fibrosis development. In particular TGF- β has emerged as key factor in fibrosis development⁹⁻¹². Additionally, megakaryocytes secrete CXCL4, which induces myofibroblastic differentiation of mesenchymal stem cells, further promoting fibrosis^{13,14}. By reducing the number of megakaryocytes, IFM-2384 likely decreases the release of these pro-fibrotic factors, leading to a reduction in fibrosis.

8) An important issue with current MPN treatments, specifically with Ruxolitinib, is the development of severe cytopenias, which ultimately results in patients coming off of the drug. Have the authors tested the synergistic effect of Rux and IFM2384 on human cells/HSPCs (ie. CFU assays)?

We have not yet tested the combination of IFM-2384 and ruxolitinib in human HSPCs, such as through CFU assays.

It is important to note, however, that one of the main effects of IFM-2384 observed in our studies was a reduction in platelet numbers, which could be particularly relevant in MPNs where excessive platelet production is a central issue, such as in essential thrombocythemia (ET) and prefibrotic primary myelofibrosis (prePMF). For this reason, we wrote in the Discussion section of the manuscript that “[...] our data confirm NLRP3 as a therapeutic target in patients with MPN, particularly those in whom excessive platelet

production is of primary concern, such as in ET and prefibrotic PMF.” Given this effect of IFM-2384 on platelet reduction, we would not consider it to be suitable for patients with MPN-related cytopenias, including those who experience cytopenias due to ruxolitinib treatment.

9) Which cell populations specifically drive this NLRP3 expression/disease progression? Is this primarily neutrophils?

To address cell-type specific inflammasome activation, we performed intracellular pro-IL-1 β staining in both mature and immature bone marrow cells from *Jak2^{VF}* and *Vav-Cre* mice.

Interestingly, we found that JAK2V617F-positive neutrophils did not display higher levels of pro-IL-1 β compared to their *Vav-Cre* counterparts, arguing against the notion that neutrophils are the primary drivers of NLRP3-dependent disease progression in our model.

Surprisingly, among mature cell populations, we observed a considerable increase in pro-IL-1 β production in B cells when JAK2V617F was present. This suggests that B cells may contribute to the NLRP3-driven inflammatory response in MPN.

Among immature cell populations, we found that JAK2V617F elevated pro-IL-1 β levels across all examined populations, with statistical significance reached in CMPs, MEPs, and MkPs. Staining with 5-FAM-YVAD-FMK confirmed higher caspase activity in JAK2V617F mutant CMPs, MEPs, and MkPs compared to *Vav-Cre* cells.

These findings suggest that a broad range of myeloid and lymphoid cell populations, including B cells and several progenitor populations, contribute to the NLRP3-mediated inflammatory cascade in the context of JAK2V617F-induced MPN.

The data is shown in Figure S3.

Minor Comments:

The figures are generally well designed, but please include individual data points to provide more transparency.

We have revised the graphs as requested. While this has made them slightly more cluttered, we agree that they are now more informative.

Line 230 – typo "capase-1"

We have corrected this in the revised manuscript.

Reviewer #2 (Remarks to the Author): with expertise in inflammasome, haematopoiesis

Earlier studies (Refs 9,10) have shown that inhibition of IL-1b or IL-1 signaling reduced platelet counts and features of myelofibrosis in Jak2VF mice. The present work extends these findings by showing that these changes are being driven in major part by NLRP3 inflammasome activation in chow fed mice. They also provide new mechanistic insights into the role of the NLRP3 inflammasome in driving the short route from HSCs to MkPs, although there is limited insight into whether this is a direct effect of IL-1b in the LSK population and if so the changes that are involved. Importantly, they show that an NLRP3 inhibitor reduces platelet counts and features of myelofibrosis. Overall, the findings are novel and interesting, the studies are carefully performed and the paper is well written.

We greatly appreciate that the reviewer recognizes the novelty and rigor of our findings and positively assesses the quality of the study and writing.

1) Ref 34 already reported increased NLRP3 inflammasome activation in murine and human JAK2VF cells, and showed increased mitochondrial ROS generation as a potential underlying mechanism. This does not detract from the present report but needs to be acknowledged.

In the revised manuscript we have updated the Introduction to acknowledge that Fidler et al.¹⁵ demonstrated increased NLRP3 inflammasome activation in JAK2V617F mutant murine BMDMs and human macrophages derived from induced pluripotent stem cells.

Regarding the possible role of mitochondrial ROS in enhanced NLRP3 inflammasome activation, based on data by Bauernfeind et al.¹⁶ we hypothesize that ROS may be a priming factor but does not activate NLRP3. We therefore already noted in the Discussion of the original manuscript that “ROS inhibitors appear to rather block priming than activation”.

2) The evidence for pyroptosis of HSCs is based on indirect measurements. Can the authors use an antibody to the N-terminal fragment of GSD to confirm these findings?

The reviewer is correct that our evidence for pyroptosis of HSPCs is based on indirect measurements. We used 5-FAM-YVAD-FMK staining and low forward scatter to identify cells undergoing pyroptosis.

Following the reviewer's suggestion, we attempted to stain for cleaved Gasdermin D using antibodies targeting Cleaved Gasdermin D Asp276 (Cell Signaling #10137) and Cleaved Gasdermin D Gly277 (Cell Signaling #34667). Unfortunately, we encountered significant issues with nonspecific staining using both antibodies, which precluded obtaining reliable and interpretable results.

To ensure transparency, we have included this limitation in the revised manuscript. While we acknowledge the value of direct confirmation, the current data remain consistent with pyroptosis based on the methods we were able to validate.

3) What is the mechanism by which NLRP3 influences the HSC-Mkp axis?

To answer this question, we have injected WT mice with IL-1 β and monitored CD48^{lo} MkPs (directly connected to HSCs) and CD48^{hi} MkPs (derived from CMPs). We found that IL-1 β increased the frequency of CD48^{lo} MkPs while reducing the frequency of CD48^{hi} MkPs among the total Mkp population in the bone marrow. These findings suggest NLRP3 promotes production of MkPs via the short route through IL-1 β . Our data are in line with those from Rai et al.¹⁷ and Rahman et al.¹⁸ who showed that excessive platelet production in JAK2V617F mutant MPN depends on IL-1 β .

The data are shown in Figure 6i.

4) Experiments done with the NLRP3 inhibitor (IFM) lack a WT control group. In view of the therapeutic potential it would be important to at least show no effect on platelets in WT controls.

While we understand the concern, we respectfully disagree with the suggestion that demonstrating an effect of NLRP3 inhibitors on platelets in WT controls would diminish their therapeutic potential in MPN. Several approved therapies for MPN clearly affect platelets in non-MPN individuals without compromising their clinical utility. For instance, pegylated interferon has been shown to reduce platelets in non-MPN individuals that receive the drug for viral hepatitis^{19,20} but is still successfully used in the treatment of essential thrombocythemia and polycythemia vera. Similarly, ruxolitinib and hydroxyurea (HU) reduce platelet counts in non-MPN patients, including those with graft-versus-host disease (GvHD)²¹ and sickle cell anemia²², respectively, yet remain valuable treatments in MPN. Therefore, the effect of NLRP3 inhibition on platelets in WT mice does in no way preclude its therapeutic potential in MPN.

We are in the process of investigating the role of NLRP3 in WT hematopoiesis. However, this is not straightforward, as it is influenced by the background level of NLRP3 activation. We alluded to this complexity in the Limitations section of the Discussion, where we

stated: “We also did not investigate the effect of the NLRP3 inflammasome on WT hematopoiesis. Given that a Western-type, calorically rich diet induces Nlrp3-dependent inflammation, we anticipate that any such analysis would need to consider dietary conditions.” Additionally, the hygiene level in the animal facility may also play a role in modulating NLRP3 activation and could influence hematopoiesis.

5) Line 151. By “real world conditions” the authors mean uncontrolled observational conditions. Previous reports have shown that Rux decreases IL-18 and increases LDL levels under conditions where the patient is compared before and after treatment.

The reviewer is correct that registry data are inherently observational and non-interventional. As we mentioned in response to Reviewer #1's Question 1, such data are subject to variability due to numerous factors, including drug dose, treatment line, risk group, and other clinical variables (please see above). Therefore, we believe that the term "real-world conditions" is appropriately used to reflect these characteristics.

However, we appreciate the reviewer’s point and would like to clarify that we are aware of studies comparing serum cytokine levels in primary myelofibrosis patients before and after ruxolitinib treatment. Specifically, data from a phase 1-2 trial demonstrate that ruxolitinib can influence some serum cytokine levels within 28 days of treatment. **We included this reference in the revised manuscript to provide additional context²³.**

Reviewer #3 (Remarks to the Author): with expertise in myeloproliferative neoplasms, haematopoiesis

Koerber et al., in this manuscript, explored the roles of NLRP3 inflammasome in JAK2V617F MPN disease development and potential therapy. Through cytokine profiling in clinical samples, they found that IL-1 β and IL-18, which were closely associated with the activation of NLRP3 inflammasome, were significantly higher in MPN patients than in healthy individuals. Using JAK2V617F knock-in mice, the authors found that loss of NLRP3 attenuated MPN phenotypes, including platelet and white blood cell counts in peripheral blood and hematopoietic stem and progenitor cell (HSPC) subpopulation numbers in the bone marrow via decreased IL-1 β and IL-18 secretion. Furthermore, the authors found that NLRP3 regulates thrombopoiesis and promotes fibrosis and splenomegaly in JAK2V617F knock-in mice. Finally, blocking NLRP3 with the novel oral inhibitor was found to have potential therapeutic effectiveness in treating MPN. The authors claimed that NLRP3 is critical for MPN development, and its inhibition represents a new therapeutic intervention for MPN patients. My major comments are listed below:

1. NLRP3/Caspase 1/IL-1 β pathway is well-known in many inflammation-mediated diseases, including myeloproliferative neoplasms (Shivam Rai et al., Nature Communications, 2022, Mohammed Ferdous-Ur Rahman et al., Nature Communications, 2022). NLRP3's roles shown in this paper are very similar to those of IL-1 β -IL1R1 in those two published papers. Comparing NLRP3 and IL-1 β side by side in many assays to show NLRP3's unique roles may strengthen the novelty of this study.

We referenced both of the studies cited by the reviewer in the manuscript. In fact, Radek Skoda, the last author of one of these studies, kindly provided feedback on our manuscript.

In the Discussion, we carefully consider possible mechanisms by which NLRP3 may exert its effects in MPNs, including through IL-1 β /IL-1R1 signaling (starting at the sentence: "NLRP3 might promote MPN development through several mechanisms, including maturation of IL-1 β and IL-18, initiation of pyroptosis by cleaving the pore-forming cell death executor gasdermin D, and degradation of the hematopoietic lineage specifying transcription factor GATA1.>").

We fully acknowledge that the phenotype observed in our model is likely at least partly due to IL-1 β . Importantly, our study shows that serum concentrations of IL-1 β and IL-18 in JAK2V617F mice lacking NLRP3 are reduced to levels comparable to those in WT mice, suggesting that NLRP3 is upstream of these cytokines and is the dominant inflammasome in MPN (and not e.g. AIM2). We also explicitly clarify what NLRP3 does **not** do, in contrast to what has been implicated in the literature. For example, we demonstrate that NLRP3-dependent reduction of hemoglobin through GATA1 degradation does not appear to play a role in our model, and that NLRP3 does not induce cytopenias through pyroptosis, as suggested in models of MDS.

These findings help mitigate concerns about other potential effects of NLRP3 inhibition that could affect its therapeutic utility. Overall, we believe our data are complementary to the existing literature rather than overlapping. We do not think that directly replicating IL-1 β or IL-1R1 inhibition experiments would add significant value to our narrative.

2) Can the authors perform a competitive bone marrow transplant experiment to determine whether NLRP3 inhibitor affects JAK2V617F allele burden? It would be interesting to explore the combination of Ruxolitinib and NLRP3 inhibitor. Figure 1g showed that Ruxolitinib did not affect IL-1 β and IL-18 secretion. Combination with NLRP3 inhibitor may avoid Ruxolitinib resistance.

Ruxolitinib resistance is indeed a major clinical challenge in myelofibrosis. Resistance to ruxolitinib is often associated with cytopenias, which require dose reductions and may limit the therapeutic efficacy for spleen and symptom control. However, we are unaware of mouse models that faithfully replicate this phenotype frequently observed in

myelofibrosis patients. In most MPN models, cytopenias only develop at a very late stage. In the case of cytopenias we believe NLRP3 inhibitors not to be suitable combination partners for ruxolitinib. As we mentioned in response to Reviewer #1's Question 8, one of the key effects of the NLRP3 inhibitor IFM-2384 in our study was a reduction in platelet numbers, limiting its potential use in patients already suffering from cytopenias. Therefore, we see the therapeutic potential of NLRP3 inhibition more in earlier stages in essential thrombocythemia (ET) and prefibrotic primary myelofibrosis (prePMF), rather than in later-stage cytopenic myelofibrosis patients. It is also important to note that complete (genetic) inhibition of NLRP3 already reduces IL-1 β and IL-18 to wild-type levels (former Figure 2d, now Figure 3a). Therefore, instead of directly combining NLRP3 inhibitors with ruxolitinib, we believe it would be more prudent to first optimize the dosing regimen of the NLRP3 inhibitor to better understand its potential before considering combinations.

We also agree that testing whether NLRP3 inhibition can induce a molecular response is an interesting idea. However, given the time required to obtain the necessary regulatory approvals, we estimate that such experiments would take 1.5 to 2 years to complete. Considering the substantial dataset already presented in our manuscript, we do not believe that further investigation into the molecular response would justify such a long delay in publication at this stage.

3) The authors showed that the loss of NLRP3 impairs platelet production via the short route of HSCs to MkPs. However, how NLRP3 regulates thrombopoiesis is unclear. RNA-seq or proteomic assay may help provide some clues.

As detailed in the answer to Reviewer #2's Question 3, we have now conducted experiments that indicate NLRP3 promotes production of MkPs via the short route through IL-1 β (please see above).

Nevertheless, we followed the reviewer's recommendation and performed 3'-RNAseq on HSCs and MkPs sorted from *Jak2^{VF}* and *Jak2^{VF};Nlrp3^{-/-}* mice. The analysis revealed that deleting *Nlrp3* enhances the expression of an oxidative phosphorylation (OXPHOS) signature in these cell types. We validated this metabolic shift using a biochemical assay measuring OXPHOS and glycolysis. Inhibiting OXPHOS with oligomycin demonstrated that mitochondrial ATP production was higher in *Jak2^{VF};Nlrp3^{-/-}* than in *Jak2^{VF}* bone marrow cells. These findings are consistent with reports that NLRP3 promotes glycolysis via IL-1 β and PFKFB3²⁴. While speculative, it is possible that this shift away from glycolysis in *Jak2^{VF};Nlrp3^{-/-}* mice may reduce the thromboembolic risk because glycolysis supports platelet activation^{25,26}.

The results are presented in Figure S4.

4) The authors showed in Figure 4a that CMPs, GMPs, MEPs, and MkPs were diminished in Jak2VF Nlrp3^{-/-} BM relative to Jak2VF BM mice. However, these figures were not shown. It is also unclear whether other HSPC subpopulations were affected.

We thank the reviewer for pointing out the potential confusion in the wording. We apologize for the lack of clarity. In the manuscript, we had written: "The size of the LS-K cell population in the bone marrow containing CMPs (common myeloid progenitor), GMPs (granulocyte-monocyte progenitor), MEPs (megakaryocytic-erythroid progenitor), and MkPs (megakaryocyte progenitor) was diminished in Jak2VF;Nlrp3^{-/-} BM relative to Jak2VF BM mice and comparable to that in WT BM mice (Fig. 4a and Fig. S3)." This may have inadvertently suggested that we were showing these subpopulations individually. To clarify, Figure 5a (former Figure 4a) shows the LS-K cell population as a whole, which includes CMPs, GMPs, MEPs, and MkPs. The individual subpopulations are not shown, except for MkPs in Figure 6c.

To avoid confusion, we have revised the wording to: "The size of the LS-K cell population in the bone marrow was diminished in Jak2VF;Nlrp3^{-/-} BM relative to Jak2VF BM mice and comparable to that in WT BM mice (Fig. 5a and gating schematic in Fig. S2)."

Regarding other HSPC subpopulations, Figure 5a (former Figure 4a) does indeed show HPC-1 (MPP-3), HPC-2 (MPP-2), MPP, and HSC populations. We hope this clarification resolves any ambiguity.

5) Figure 6c is not convincing. The Jak2VF Nlrp3^{-/-} group had 9 mice in the body weight panel, and there were 8 mice in the Jak2VF Nlrp3^{-/-} group in the relative spleen weight panel. What would the statistics be like when all the mice are included? A similar case was found in Figure 6b. These figures are not convincing and should be revised.

We understand the concern regarding the number of mice used in different panels and appreciate the opportunity to clarify these points.

Regarding Figure 6b (now Figure 7b), the bone marrow fibrosis and spleen fibrosis results were derived from independent cohorts. Initially, we only planned to assess bone marrow fibrosis, but later decided to include spleen fibrosis as an additional parameter. We did not discard any data points; rather, the decision to include spleen fibrosis was made after the bone marrow fibrosis cohort had been completed.

For Figure 6c (now Figure 7d), we acknowledge the discrepancy in the number of Jak2VF; Nlrp3^{-/-} mice between the body weight (9 mice) and relative spleen weight (8 mice) panels. The data presented for both body weight and relative spleen weight are from the same set of animals. The discrepancy arose because, unfortunately, we failed to document the spleen weight for one mouse. It is important to note that the mice included in this analysis were born on different dates and were sacrificed over the course of more than a

year. In some cases, only a single mouse was sacrificed at a time, which led to the oversight. We understand that the missing data point may cause concern about the consistency of the data, but we believe that the scatter plots clearly demonstrate that all data points are presented transparently and that no data have been hidden. **In response to this comment, we have added additional mice to the analysis comparing Jak2VF and Jak2VF; Nlrp3^{-/-} mice for relative spleen weight to further support our findings.** We hope this will provide additional confidence in the comparison.

Lastly, as per Reviewer #1's Minor Comment 1, we have revised all figures in the manuscript to show individual data points for greater transparency.

Reviewer #4 (Remarks to the Author): with expertise in myeloproliferative Neoplasms, inflammasome

The work by Koerber and colleagues is focused on the role of inflammasome activation in MPN patients and JAK2VF murine cells and explores the possible benefit of inhibiting this pathway by using genetic and pharmacologic approaches. They demonstrate elevated levels of inflammasome-processed cytokines IL-1 β and IL-18 in patient and murine serum, which are reduced in the absence of NLRP3. Inflammasome activation is suggested by increased ASC speck formation in patient cells and the release of mature IL-1 β from JAK2VF macrophages following NLRP3 stimulation with LPS+Nigericin. NLRP3 ablation or pharmacologic inhibition reduced platelet, and to a lesser extent, leukocyte counts with no changes in hemoglobin levels, reduced HSCP and MK numbers, BM and spleen fibrosis and splenomegaly, pointing to NLRP3 as a therapeutic target in MPN, which could be of potential clinical interest. The work is well performed and novel. However, insight into the signals leading to NLRP3 activation as well as the mechanism underlying the effect of NLRP3 on the myeloproliferative phenotype are limited, which diminishes the impact of the findings.

We thank the reviewer for the thoughtful and constructive feedback. We very much appreciate the fact that she/he considers the study to be of high quality and that it contains novel insights. We will elaborate on the mechanism that leads to NLRP3 activation in our responses below.

Comments:

1-Figure 7D seems to be missing. Please include it.

Figure 7D was indeed included in the original manuscript, but it was placed at the top right corner of the figure to optimize the use of space. However, this placement may cause it to be overlooked.

In the revised manuscript, the former Figure 7 is now Figure 8. Panel 7d has been moved to panel 8e where it can be found in the expected location, between panels 8d and 8f.

2) Evidence for enhanced inflammasome activation in patient cells (mild increase in ASC specks by flow cytometry) should be reinforced by additional experiments, such as for example measurement of caspase 1 activation or further confirmation of the presence of ASC specks by another method. Besides, it would be relevant to further demonstrate the specific participation of NLRP3, for example by showing colocalization of NLRP3 in ASC specks by immunofluorescence and/or blockade of ASC speck formation by using a specific NLRP3 inhibitor. Besides, the authors state that ASC specks were increased in peripheral blood "myeloid" cells: are these neutrophils? monocytes? myeloid precursors? how were these cells identified?

We appreciate the reviewer's insightful suggestions and have addressed the comments as follows:

Evidence for Inflammasome Activation:

In unstimulated PBMCs, we observed a 3-fold increase in the percentage of ASC speck-positive cells, rising from 0.03% to 0.09%. We acknowledge that the absolute percentage of ASC speck-positive cells is relatively low, but it is important to note that inflammasome activation generally leads to cell death, and thus, these cells are cleared from circulation. This explains the low levels we detect, which are consistent with the dynamic nature of inflammasome activation and the transient nature of ASC speck formation.

We further highlight that even small increases in inflammasome activation may have significant biological effects when sustained over a long period of time. This is particularly relevant in chronic conditions like MPNs.

In addition to ASC speck formation, we also present data showing increased levels of IL-1 β and IL-18 in patient samples. Thus, we already included several read-outs for inflammasome activation, which support our conclusion of enhanced inflammasome activation in these patients.

Participation of NLRP3 in ASC Speck Formation:

In response to the reviewer's suggestion, we have now performed additional experiments to demonstrate the involvement of NLRP3 in ASC speck formation. Specifically, we analyzed co-localization of NLRP3 and ASC specks in PBMCs from MPN patients by flow cytometry. The results show clear co-staining of ASC specks with NLRP3, indicating the formation of NLRP3-ASC complexes during inflammasome activation.

Furthermore, we tested the impact of NLRP3 inhibition on ASC speck formation by treating patient PBMCs with the specific NLRP3 inhibitor MCC950. Our findings demonstrate that MCC950 treatment significantly reduced the percentage of ASC speck-positive cells, further supporting the contribution of NLRP3 to the observed inflammasome activation in MPN patients.

These findings are now presented in Figure 2e-g.

Cell identity of ASC speck-positive cells:

Regarding the identity of the ASC speck-positive cells in human blood samples, we used Ficoll gradient centrifugation to isolate PBMCs thereby removing granulocytes. In the flow cytometric analysis. We co-stained for the surface markers CD3, CD19, CD14 and HLA-DR. The myeloid cell fraction in PBMCs mostly consisted of monocytes.

In response to Reviewer #1's Question 9, we have studied the specific cell types, both mature and immature, that produce IL-1 β in Jak2VF mice (see above for detailed results). This analysis includes neutrophils and shows that IL-1 β is not elevated in Jak2VF neutrophils.

3) The mechanism underlying inflammasome hyperactivation in MPN is not entirely clear. The authors suggest it might be due to JAK2-dependent inflammasome priming. Elevated CASP1 and IL-1 β in patient cells is shown in Figure 2A. Considering the work is overall focused on NLRP3, the authors should also show NLRP3 levels in these samples and analyze whether there are differences between JAK2-positive and -negative patients. NLRP3 levels could also be assessed in the murine JAK2VF model. To this reviewer's knowledge, NLRP3 has not been described as a downstream JAK2 target.

A previous study by Zhou et al. has already demonstrated increased NLRP3 transcript levels in MPN¹. To extend these results we have now performed additional experiments to assess NLRP3 protein expression in MPN. Specifically, we analyzed NLRP3 protein levels by immunohistochemistry (IHC) on bone marrow sections from Jak2VF, WT and Nlrp3^{-/-} mice. This IHC analysis revealed a robust elevation of NLRP3 in Jak2VF mice, supporting the hypothesis that JAK2 promotes NLRP3 protein expression. The data is shown in Figure 3d and 3e. It is important to note that NLRP3 does not necessarily need to be a direct downstream target of JAK2; rather, its expression could be enhanced indirectly through the upregulation of inflammatory cytokines that induce NF- κ B. The NLRP3 gene is a direct target of NF- κ B.

4) Following on this comment, could high TNFalpha levels contribute to NLRP3 inflammasome activation in the JAK2VF model? It could be of interest to test whether NLRP3 activation is ameliorated by the use of TNF blocking antibodies or inhibitors.

TNF- α is a well-established priming factor for the NLRP3 inflammasome. While we have not specifically tested whether TNF- α contributes to NLRP3 activation in the JAK2VF mouse model, there is evidence from related studies that TNF signaling plays a role in MPN pathology.

For example, Müller et al. used a JAK2V617F knock-in mouse model and demonstrated that inhibition of TNFR2 led to a reduction in white blood cell counts, though it did not improve hematocrit or splenomegaly²⁷. In contrast, blockade of TNFR1 reduced hematocrit and splenomegaly, suggesting a role for TNF receptors in MPN-related pathology.

Moreover, TNF- α blockade has been tested in clinical settings²⁸. In a study using the fusion protein etanercept, which binds and neutralizes TNF- α , 60% of MPN patients (12 out of 22) showed improvement in constitutional symptoms, and 20% of patients (4 out of 22) exhibited improvement in cytopenia or spleen size. These findings suggest that TNF- α signaling could be a key contributor to disease progression and may impact inflammasome activation.

5) JAKVF NLRP3-/- mice show lower platelet/leukocyte counts vs. JAKVF and WT mice. Blood counts of NLRP3-/- mice should be shown for comparison to exclude this is due to the effect of NLRP3 ablation on normal hematopoiesis.

A similar point was raised by Reviewer #2. We kindly refer the reviewer to our response to Reviewer #2's Question 4 for further clarification.

6) The authors state that NLRP3 ablation reduces platelet counts at least in part by inhibiting the direct thrombopoiesis pathway, as shown by lower levels of CD48-low MKs. They suggest this feature could be due to the reduction in IL-1 β . Does IL-1 β regulate the direct thrombopoiesis pathway? Further experiments should be performed to support this statement, for example evaluating this pathway in JAK2VF with IL-1 β ablation or pharmacologic IL-1 β inhibition; alternatively the effect of recombinant IL-1 β in wild-type mice would also be informative.

We agree that a clearer definition of the mechanism by which NLRP3 contributes to the direct thrombopoiesis pathway is needed. A similar point was raised by Reviewer #2. To avoid repetition and keep the letter concise, we would ask the reviewer to please see our response to Reviewer #2's Question 3.

7) *The reference list does not comprehensively cover the available literature: a) Elevation of inflammasome-processed cytokines IL-1 β (PMID: 36100613, PMID: 36100596) and IL-18 (PMID: 38035089) has been previously described in plasma/serum from MPN patients. The authors should refer to these previous findings. b) Enhanced NLRP3 inflammasome activation was previously shown in LPS and Nigericin-stimulated monocytes from patients with Myelofibrosis (PMID: 38035089). Please acknowledge these findings in the manuscript.*

We thank the reviewer for highlighting these important references. However, the first two (PMID: 36100613 and PMID: 36100596) were already cited in the Introduction of the manuscript, where we noted that “Recent work has established IL-1 β as a major regulator of inflammation in the aging hematopoietic stem cell niche and in MPN.” **We have now also incorporated the third reference (PMID: 38035089) into the Introduction.**

Minor comments

Please explain why TNF α is reduced by NLRP3 deletion/inhibition in the JAKVF model (Figure 2D and Figure 7A)?

The reduction in TNF- α following NLRP3 deletion/inhibition in the JAKVF model (now Figures 3a and 8a) can be explained by the role of the NLRP3 inflammasome in regulating IL-1 β . NLRP3 activation leads to IL-1 β release, which subsequently induces NF- κ B signaling. Canonical NF- κ B activation then promotes the production of TNF- α . This can create a feedback loop in which TNF- α further activates NF- κ B signaling, which primes the NLRP3 inflammasome.

Please detail how were MkP identified (Figure 5C).

In the original manuscript, we already included a gating schematic in Supplemental Figure S3 (now S2) showing how we identified different progenitor cells. MkPs were identified as Lin⁻CD117⁺Sca-1⁻CD150⁺CD41⁺ cells.

It is surprising that differences in TPO levels in the different animal groups were not significant, as data look quite different (Figure 5D).

While there were trends, the statistical analysis did not reveal significant differences in TPO levels across the different animal groups. These are the adjusted p-values:

Jak2VF; Nlrp3^{-/-} vs Jak2VF: 0.3561

Jak2VF; Nlrp3-/- vs WT: 0.4879

Jak2VF vs WT: 0.1354

As requested by Reviewer #1, we revised the manuscript to include individual data points in the bar graphs to improve transparency. Initially, we were concerned that this might lead to cluttered diagrams, but Figure 6e (former Figure 5d) is a good example why this approach makes sense. The mean values in Figure 6e were considerably influenced by single data points.

In Figure 1E, 2A and 2B, please detail which MPN patients were included in each assay (number of ET, PV and PMF patients).

In the former Figure 1e (now Figure 2b), the MPN patient population includes 19 with PV, 7 with ET, and 10 with PMF.

In the former Figure 2a (now Figure 1d), the “Monocytes” groups consist of 20 PV, 8 ET, and 24 PMF patients, while the “Monocyte-depleted PBMCs” groups include 4 PV, 5 ET, and 10–11 PMF patients.

The former Figure 2b (now Figure 1e) is based on the 3'-TARGET-seq dataset GSE122198, as indicated in the figure legend. This analysis includes single cells from 8 JAK2V617F-positive myelofibrosis patients (4 PMF, 3 post-PV MF, and 1 post-ET MF). It shows enrichment scores for a gene set related to Inflammasome Priming (IL1B, IL18, GSDMD, PYCARD, CASP1) in single HSPCs that are either JAK2 WT or JAK2V617F mutant.

We have added this information to the figure legends in the revised manuscript.

Figure 1 A data are shown in two separate sets. Please explain how were these patients divided in the figure legend.

The heatmap was created by applying k-means clustering to both the rows (cytokines) and columns (individuals), grouping similar cytokines and similar individuals together. We chose to use 2 clusters (k=2) for both rows and columns, resulting in 2x2 clusters. While it is possible to perform a more detailed partitioning with more groups, we decided against it to avoid over-complicating the data.

Figure 2 A and B. Please provide captions for the colour code in the graphs (do gray bars represent HC and red bars, patients?)

We apologize for the oversight in the figure labeling. The reviewer is correct: the gray bars represented healthy controls (HC), while the red bars represented MPN patients. **We have revised the figure to include these labels (now Figure 1d).**

References:

1. Zhou, Y. *et al.* Genetic polymorphisms and expression of NLRP3 inflammasome-related genes are associated with Philadelphia chromosome-negative myeloproliferative neoplasms. *Human Immunology* **81**, 606–613 (2020).
2. Brydges, S. D. *et al.* Inflammasome-Mediated Disease Animal Models Reveal Roles for Innate but Not Adaptive Immunity. *Immunity* **30**, 875–887 (2009).
3. McGeough, M. D. *et al.* TNF regulates transcription of NLRP3 inflammasome components and inflammatory molecules in cryopyrinopathies. *Journal of Clinical Investigation* **127**, 4488–4497 (2017).
4. Beaulieu, L. M. *et al.* Interleukin 1 Receptor 1 and Interleukin 1 β Regulate Megakaryocyte Maturation, Platelet Activation, and Transcript Profile During Inflammation in Mice and Humans. *ATVB* **34**, 552–564 (2014).
5. Kuter, D. & Rosenberg, R. The reciprocal relationship of thrombopoietin (c-Mpl ligand) to changes in the platelet mass during busulfan-induced thrombocytopenia in the rabbit. *Blood* **85**, 2720–2730 (1995).
6. Lenkiewicz, A. M. *et al.* The Nlrp3 Inflammasome Orchestrates Mobilization of Bone Marrow-Residing Stem Cells into Peripheral Blood. *Stem Cell Rev and Rep* **15**, 391–403 (2019).
7. Adamiak, M. *et al.* Nlrp3 Inflammasome Signaling Regulates the Homing and Engraftment of Hematopoietic Stem Cells (HSPCs) by Enhancing Incorporation of CXCR4 Receptor into Membrane Lipid Rafts. *Stem Cell Rev and Rep* **16**, 954–967 (2020).
8. Luo, H. *et al.* Mitochondrial Stress-Initiated Aberrant Activation of the NLRP3 Inflammasome Regulates the Functional Deterioration of Hematopoietic Stem Cell Aging. *Cell Reports* **26**, 945-954.e4 (2019).
9. Chagraoui, H. *et al.* Prominent role of TGF- β 1 in thrombopoietin-induced myelofibrosis in mice. *Blood* **100**, 3495–3503 (2002).
10. Gastinne, T. *et al.* Adenoviral-mediated TGF- β 1 inhibition in a mouse model of myelofibrosis inhibit bone marrow fibrosis development. *Experimental Hematology* **35**, 64–74 (2007).
11. Zingariello, M. *et al.* Characterization of the TGF- β 1 signaling abnormalities in the Gata1^{low} mouse model of myelofibrosis. *Blood* **121**, 3345–3363 (2013).
12. Ceglia, I. *et al.* Preclinical rationale for TGF- β inhibition as a therapeutic target for the treatment of myelofibrosis. *Experimental Hematology* **44**, 1138-1155.e4 (2016).
13. Schneider, R. K. *et al.* Gli1 + Mesenchymal Stromal Cells Are a Key Driver of Bone Marrow Fibrosis and an Important Cellular Therapeutic Target. *Cell Stem Cell* **20**, 785-800.e8 (2017).

14. Gleitz, H. F. E. *et al.* Increased CXCL4 expression in hematopoietic cells links inflammation and progression of bone marrow fibrosis in MPN. *Blood* **136**, 2051–2064 (2020).
15. Fidler, T. P. *et al.* The AIM2 inflammasome exacerbates atherosclerosis in clonal haematopoiesis. *Nature* **592**, 296–301 (2021).
16. Bauernfeind, F. *et al.* Cutting Edge: Reactive Oxygen Species Inhibitors Block Priming, but Not Activation, of the NLRP3 Inflammasome. *The Journal of Immunology* **187**, 613–617 (2011).
17. Rai, S. *et al.* Inhibition of interleukin-1 β reduces myelofibrosis and osteosclerosis in mice with JAK2-V617F driven myeloproliferative neoplasm. *Nat Commun* **13**, 5346 (2022).
18. Rahman, M. F.-U. *et al.* Interleukin-1 contributes to clonal expansion and progression of bone marrow fibrosis in JAK2V617F-induced myeloproliferative neoplasm. *Nat Commun* **13**, 5347 (2022).
19. Yamane, A. *et al.* Interferon- α 2b-induced thrombocytopenia is caused by inhibition of platelet production but not proliferation and endomitosis in human megakaryocytes. *Blood* **112**, 542–550 (2008).
20. Sata, M. *et al.* Mechanisms of thrombocytopenia induced by interferon therapy for chronic hepatitis B. *J Gastroenterol* **32**, 206–210 (1997).
21. Zeiser, R. *et al.* Ruxolitinib for Glucocorticoid-Refractory Acute Graft-versus-Host Disease. *N Engl J Med* **382**, 1800–1810 (2020).
22. Charache, S. *et al.* Effect of Hydroxyurea on the Frequency of Painful Crises in Sickle Cell Anemia. *N Engl J Med* **332**, 1317–1322 (1995).
23. Verstovsek, S. *et al.* Safety and Efficacy of INCB018424, a JAK1 and JAK2 Inhibitor, in Myelofibrosis. *N Engl J Med* **363**, 1117–1127 (2010).
24. Finucane, O. M., Sugrue, J., Rubio-Araiz, A., Guillot-Sestier, M.-V. & Lynch, M. A. The NLRP3 inflammasome modulates glycolysis by increasing PFKFB3 in an IL-1 β -dependent manner in macrophages. *Sci Rep* **9**, 4034 (2019).
25. Ghatge, M., Flora, G. D., Nayak, M. K. & Chauhan, A. K. Platelet Metabolic Profiling Reveals Glycolytic and 1-Carbon Metabolites Are Essential for GP VI-Stimulated Human Platelets—Brief Report. *ATVB* **44**, 409–416 (2024).
26. Flora, G. D. *et al.* Mitochondrial pyruvate dehydrogenase kinases contribute to platelet function and thrombosis in mice by regulating aerobic glycolysis. *Blood Advances* **7**, 2347–2359 (2023).
27. Müller, P. *et al.* Anti-inflammatory treatment in MPN: targeting TNFR1 and TNFR2 in JAK2-V617F-induced disease. *Blood Advances* **5**, 5349–5359 (2021).
28. Steensma, D. P., Mesa, R. A., Li, C.-Y., Gray, L. & Tefferi, A. Etanercept, a soluble tumor necrosis factor receptor, palliates constitutional symptoms in patients with myelofibrosis with myeloid metaplasia: results of a pilot study. *Blood* **99**, 2252–2254 (2002).

Reviewer #3 expresses doubt regarding the novelty of our findings, stating: “The role of NLRP3 in inflammatory diseases is well established, and its involvement in MPNs has been previously reported.”

Of course, we agree that NLRP3 is a well-known player in various inflammatory conditions, as we discuss in the Introduction with reference to e.g. cryopyrin-associated periodic syndrome, type II diabetes, Alzheimer’s disease, atherosclerosis, and gout. We strongly disagree, however, with the assertion that its role in myeloproliferative neoplasms (MPNs) has already been reported. Previous studies have shown increased expression of NLRP3 in MPNs and enhanced responsiveness of JAK2V617F-mutant cells to NLRP3 stimuli. We cite these works in our Introduction. None of these studies, however, have investigated the functional role of NLRP3 in MPN, neither its global impact on disease nor its specific contributions to the phenotype. To do so requires the genetic or pharmacological inhibition of NLRP3 in a validated mouse model of MPN. We used the *Vav-Cre;Jak2V617F/+* mouse model, despite the logistical challenges it presents (i.e. long disease development period and the low yield of target genotype mice). This model enabled us to demonstrate that the production of the inflammasome-associated cytokines IL-1 β and IL-18 in MPN is entirely dependent on NLRP3 and that consequently other inflammasome sensors such as AIM2 and NLRP1, which have also been implicated in MPN, do not appear to play a relevant role in this context. Our data further reveal a specific pathogenic function of NLRP3 in promoting excessive platelet production. We provide novel mechanistic insight by showing that NLRP3 drives platelet overproduction through a differentiation shortcut that directly connects hematopoietic stem cells (HSCs) to megakaryocyte progenitors (MkPs), adding significant depth to our study. Finally, we demonstrate that pharmacological inhibition of NLRP3 in already established disease can reverse platelet overproduction and prevent the progression to marrow fibrosis. We would like to point out that none of the other three reviewers raised concerns regarding the novelty of our findings.

Reviewer #3 also criticized that we did not adequately address whether NLRP3 inhibition can reduce the JAK2V617F allele burden and questioned why such experiments would take 1–2 years to complete.

As the reviewer correctly notes, addressing this question requires competitive transplantation experiments. However, the extended timeline is not due to the duration of the experiments themselves, but rather to the time required to obtain approval from the relevant regulatory authority, the State Agency for Nature, Environment and Consumer Protection (NRW). However, we believe that our analysis of the role of NLRP3 in MPN is already extensive. We are confident that the compelling findings presented in our manuscript, as they stand, provide a strong and well-founded rationale for advancing NLRP3 inhibitors into clinical testing for MPN.